# Knowledge-graph-based cell-cell communication inference for spatially resolved transcriptomic data with SpaTalk

Xin Shao [1,2,6], Chengyu Li [2,6], Haihong Yang [3,4,6], Xiaoyan Lu[2], Jie Liao [2], Jingyang Qian[2], Kai Wang [1], Junyun Cheng[2], Penghui Yang[2], Huajun Chen [3,4] ✉, Xiao Xu [1] ✉ & Xiaohui Fan [1,2,5] ✉

Spatially resolved transcriptomics provides genetic information in space toward elucidation of the spatial architecture in intact organs and the spatially resolved cell-cell communications mediating tissue homeostasis, development, and disease. To facilitate inference of spatially resolved cell-cell communications, we here present SpaTalk, which relies on a graph network and knowledge graph to model and score the ligand-receptor-target signaling network between spatially proximal cells by dissecting cell-type composition through a non-negative linear model and spatial mapping between single-cell transcriptomic and spatially resolved transcriptomic data. The benchmarked performance of SpaTalk on public single-cell spatial transcriptomic datasets is superior to that of existing inference methods. Then we apply SpaTalk to STARmap, Slide-seq, and 10X Visium data, revealing the in-depth communicative mechanisms underlying normal and disease tissues with spatial structure. SpaTalk can uncover spatially resolved cell-cell communications for single-cell and spot-based spatially resolved transcriptomic data universally, providing valuable insights into spatial inter-cellular tissue dynamics.

Cell–cell communications via secreting and receiving ligands frequently occur in multicellular organisms, which is a vital feature involving numerous biological processes[1]. Standard algorithms for inferring cell–cell communications mediated by ligand–receptor interactions (LRIs) primarily incorporate a database of known LRIs and single-cell transcriptomic data by delineating cell populations and their lineage relationships[2,3]. One common strategy is to integrate the abundance of ligands and receptors for the inference of signals from senders to receivers based on the premise that highly co-expressed ligands and receptors are likely to mediate inter-cellular communications[4,5]. Another strategy applies the downstream targets triggered by LRIs in receivers to enrich and score the ligand–receptor–target (LRT) signaling network[6–8]. Although single-cell transcriptomic data can provide information on the genes contributing to cell–cell communications, the spatial information of cells is inevitably lost when dissociating tissues into single cells, thereby hindering the extension of current tools to investigate cell–cell communications in tissues with spatial structure[9].

Recent technological advances in spatially resolved transcriptomics (ST) benefiting from spatial barcoding and imaging-based approaches have enabled the measurement of whole or mostly whole transcriptomes while retaining the spatial information[10], which have

[1]Key Laboratory of Integrated Oncology and Intelligent Medicine of Zhejiang Province, Department of Hepatobiliary and Pancreatic Surgery, Affiliated Hangzhou First People's Hospital, Zhejiang University School of Medicine, 310006 Hangzhou, China. [2]Pharmaceutical Informatics Institute, College of Pharmaceutical Sciences, Zhejiang University, 310058 Hangzhou, China. [3]Hangzhou Innovation Center, Zhejiang University, 310058 Hangzhou, China. [4]College of Computer Science and Technology, Zhejiang University, 310027 Hangzhou, China. [5]Future Health Laboratory, Innovation Center of Yangtze River Delta, Zhejiang University, 314100 Jiaxia, China. [6]These authors contributed equally: Xin Shao, Chengyu Li, Haihong Yang. ✉e-mail: huajunsir@zju.edu.cn; zjxu@zju.edu.cn; fanxh@zju.edu.cn

been increasingly adopted to generate useful insights in the biological and biomedical domains, with dramatically improved accuracy and reliability in the inference of spatially proximal cell–cell communications[11]. Given the space-constrained nature of juxtacrine and paracrine signaling, such spatial gene expression information is vital to understand cell–cell communications mediating tissue homeostasis, development, and disease[12,13].

Several methods have recently emerged to decode the mechanisms of cell–cell communications in space[14]. For example, Giotto utilizes preferential cell neighbors over single-cell ST datasets for each pair of cell types with an enrichment test to evaluate the likelihood of a given LRI based on proximal co-expressing cells and infer cell–cell communication in space[15]. SpaOTsc applies structured optimal transport mapping between scRNA-seq and ST data to assign a spatial position for each cell, resulting in a cell–cell distance as a transport cost to infer the ligand–receptor signaling network that mediates space-constrained cell–cell communication[16]. However, Giotto and SpaOTsc are limited to infer inter-cellular communications over single-cell ST data rather than the spot-based ST data and between paired cell types rather than paired cells. It still lacks of methods that can infer and visualize spatially resolved cell–cell communications at single-cell resolution over ST data to date, posing a great challenge for decoding spatial inter-cellular dynamics underlying disease pathology.

To address this challenge, we herein proposed SpaTalk, a spatially resolved cell–cell communication inference method by creatively integrating the principles of the ligand–receptor proximity and ligand–receptor–target (LRT) co-expression to model and score the LRT signaling network between spatially proximal cells relying on the graph network and knowledge graph approaches[17,18]. The performance of SpaTalk was evaluated on benchmarked datasets with remarkable superiority over other methods. By applying to STARmap[19], Slide-seq[20,21], and 10x Visium[22] datasets, SpaTalk revealed the in-depth communicative mechanisms underlying normal and disease tissues with spatial structure. Collectively, these results demonstrate SpaTalk as a useful and universal method that can help to uncover spatially resolved cell–cell communications for both single-cell and spot-based ST data, providing insights into the understanding of spatial inter-cellular dynamics in tissues.

## Results
### Overview of the SpaTalk method
Figure 1 provides an overview of the workflow for developing and testing SpaTalk, comprising two main components: (1) dissect the cell-type composition of ST data and (2) infer spatially resolved cell–cell communications over decomposed single-cell ST data (Fig. 1a). In the first component, the non-negative linear model (NNLM)[23–25] was applied to decode the cell-type composition for a single-cell or spot-based ST data matrix using the scRNA-seq data matrix with k cell types as the underlying reference. By incorporating Lee's multiplicative iteration algorithm and relative entropy loss[25], the model was trained with default hyperparameters until convergence, producing a weight matrix representing the optimal proportion of cell types for each cell/spot. For single-cell ST data, the cell type with the maximum weight was assigned to label each cell. For spot-based ST data, the cell types with different weights were used as the reference to project the cells from scRNA-seq data onto the spatial spot (Fig. 1b). Through random sampling and deep iteration processes, the optimal cellular combination that most resembled the spatial spot was refined to reconstruct the single-cell ST data for spot-based ST data.

The second component of SpaTalk is to infer spatially resolved cell–cell communications and downstream signal pathways. To identify possible communications among cells mediated by LRIs, the principles of ligand–receptor proximity and ligand–receptor–target (LRT) co-expression were incorporated based on a recent review[11]. In detail, the KNN algorithm is first applied to each cell in space to construct the cell graph network inspired by Giotto[15]. For a given ligand of the sender (cell type A) and a given receptor of the receiver (cell type B), the number of ligand–receptor co-expressed cell–cell pairs is obtained from the graph network by counting the 1-hop neighbor nodes of receivers for each sender. A permutation test filters and scores the significantly enriched LRIs, generating the inter-cellular score (Fig. 1c).

The knowledge graph (KG) was then introduced to model the intracellular signal propagation process from the receptor to its downstream signals, i.e., TFs and target genes. In practice, LRIs from CellTalkDB[26], pathways from Kyoto Encyclopedia of Genes and Genomes (KEGG) and Reactome, and TFs from AnimalTFDB[27] were integrated to construct the LRT-KG, wherein the weight between entities represents the co-expressed coefficient. Taking the receptor as the query node, we incorporated the random walk algorithm[28] into the LRT-KG to filter and score the downstream activated TFs and calculate the intracellular score of the LRI from senders to receivers as the LRI that mediates the cell–cell communication is supposed to activate at least one TF and its target gene in the receiver cell type and the greater number of co-expressed TFs and their target genes will lead to a higher score for a given LRI pair between the sender cell type and the receiver cell type. Inter-cellular and intracellular scores are combined to rank the LRIs that mediate spatially resolved cell–cell communications.

SpaTalk also includes numerous visualization functions to characterize the cell-type composition and spatially resolved cell–cell communications, such as the diagram of the LRI from senders to receivers in space and LRT signaling pathways, over the reconstructed single-cell ST data (Fig. 1d). Five broad ranges of different spatial technologies and corresponding representative datasets were analyzed and visualized: spot-based ST data (Slide-seq[20,21] and 10x Visium[22]) and single-cell ST data (STARmap[19], MERFISH[29], and seqFISH+[30]).

### Performance comparison of SpaTalk with other methods
The cell-type decomposition by SpaTalk is the foundation for subsequent analyses. To evaluate its performance, four single-cell ST datasets from the mouse cortex, hypothalamus, olfactory bulb (OB), and sub-ventricular zone (SVZ) were utilized (Supplementary Fig. 1a). All cells were split according to the fixed spatial distance and then merged into simulated spots as the benchmark datasets (Fig. 2a). The quality of predicted cell-type decompositions and expression profiles was evaluated by Pearson's correlation coefficient and the root mean square error (RMSE) based on the ground truth, wherein SpaTalk exhibited fantastic performance over the benchmark datasets (Supplementary Fig. 1b, c). Although the majority of existing cell-type deconvolution methods (RCTD[31], Seurat[32], SPOTlight[33], deconvSeq[34], Stereoscope[35], and Cell2location[36]) can achieve a decent correlation coefficient and low RMSE on spot deconvolution, SpaTalk outperformed these methods on most benchmark datasets with the top-ranked performance, except for the evaluation indices on the MERFISH dataset and the mean RMSE on the STARmap dataset (Fig. 2b). The MERFISH dataset only includes 155 genes, whereas the STARmap and seqFISH+ datasets cover 1020 and 10,000 genes per cell, respectively, suggesting that SpaTalk is potentially more effective for spatial data with higher gene coverage.

For the inference of cell–cell communication, we demonstrate that SpaTalk enables the identification of known LRIs in space as shown in the cases based on the STARmap and Slide-seq mouse brain ST datasets (Supplementary Fig. 2). Thus, we next compared the performance of SpaTalk with that of existing cell–cell communication inference methods with their default LRI databases (Supplementary Fig. 3a). Consequently, most methods exhibited a large fraction of overlapped predictions with the rest of the methods despite the different number of inferred cell–cell communications (Supplementary

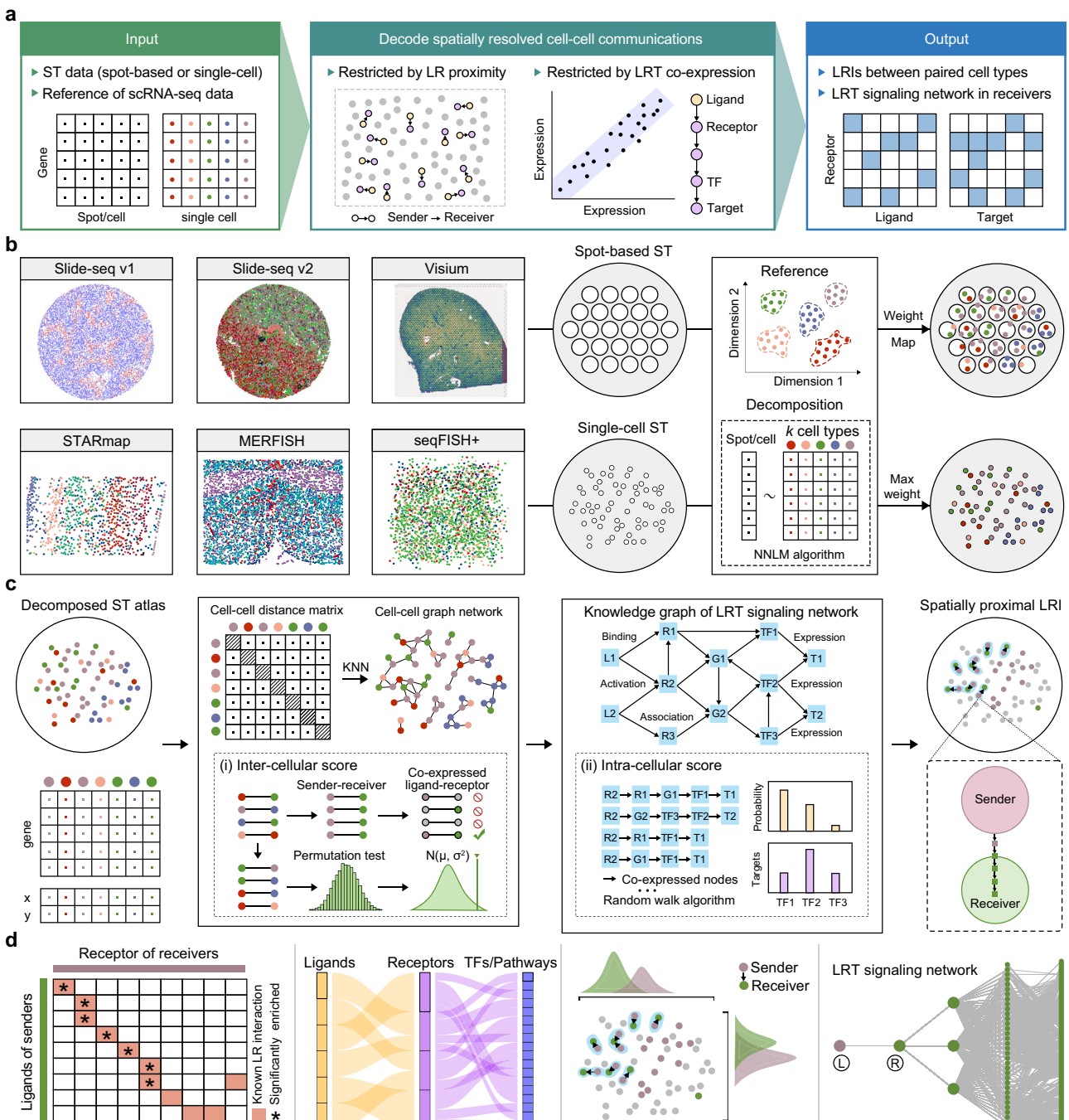

**Fig. 1 | Workflow of the SpaTalk method and visualization. a** Overview of SpaTalk, including the input, intermediate process of decoding spatially resolved cell–cell communications, and output. **b** Conceptual framework of cell-type decomposition with SpaTalk. Five different spatial technologies and datasets were selected and analyzed: spot-based ST data (Slide-seq and 10x Visium) and single-cell ST data (STARmap, MERFISH, and seqFISH+). NNLM was used to dissect the optimal proportion of cell types for the projection of cells from scRNA-seq reference data onto the spatial cells/spots, generating single-cell ST data with known cell types. **c** Schematic representation of SpaTalk to infer spatially resolved cell–cell communications mediated by LRIs. The inter-cellular and intracellular scores were obtained and combined from the cell–cell graph network and the LRT-knowledge graph (KG), respectively, by integrating the KNN, permutation test, and random walk algorithms. L, ligand; R, receptor; TF, transcription factor; T, target. **d** Visualization of spatially resolved cell–cell communications, including a heatmap, Sankey plot, and diagram of the LRI from senders to receivers in space, as well as ligand–receptor–target (LRT) signaling pathways over the reconstructed single-cell ST data.

Fig. 3b, c), indicating the reproducible inference across these methods. Regarding the inferred LRIs (Supplementary Fig. 3d), we reasoned that the spatial distances of the inferred LRI between sender-receiver pairs will be shorter than those between all cell–cell pairs and thus the inferred LRI will be more co-expressed in local space as cells that are close are more likely to signal (Fig. 2c). The one-sided Wilcoxon test

was performed to evaluate the spatial proximity significance of the inferred LRIs, and the co-expressed percentage of the LRI was calculated as the co-expression level using the cell–cell graph network. Although most LRIs inferred by other methods showed significantly closer spatial distances between sender-receiver pairs than that between all cell–cell pairs, superior performance of SpaTalk was

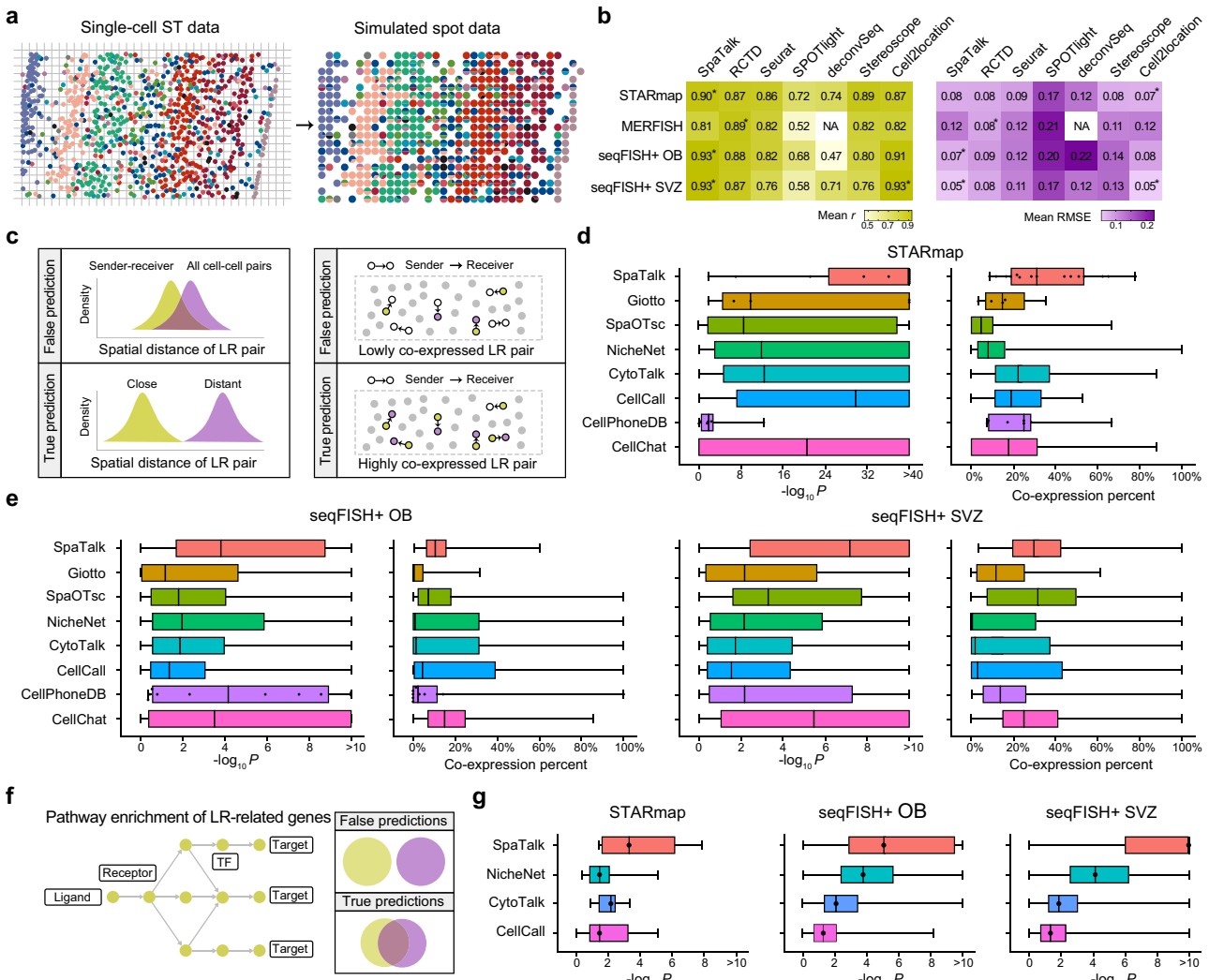

**Fig. 2 | Superior performance of SpaTalk over existing methods. a** Schematic diagram for generating simulated spot data. Cells were split according to the fixed spatial distance and then merged for the single-cell ST data with known cell types. **b** Performance comparison of SpaTalk with other existing cell-type deconvolution methods (RCTD, Seurat, SPOTlight, deconvSeq, Stereoscope, Cell2location). The asterisk represents the top-ranked method for each dataset. NA, not available. **c** Schematic illustration of the procedure and rationale for single-cell ST data to evaluate predicted LRIs that mediate spatially resolved cell−cell communications. **d** and **e** Performance comparison of SpaTalk with existing cell−cell communication inference methods (Giotto, SpaOTsc, NicheNet, CytoTalk, CellCall, CellPhoneDB, and CellChat) on the STARmap and seqFISH+ datasets. The *P*-value represents the difference of spatial distances between sender-receiver and all cell−cell pairs assessed with the Wilcoxon test. For the boxplots (minima, 25th percentile, median, 75th percentile, and maxima) from SpaTalk to CellChat, the numbers of data points for the STARmap dataset are 15, 5, 676, 537, 75, 31, 7, and 166, respectively; the

numbers of data points for the seqFISH+ OB dataset are 1,559, 38, 3,972, 1,223, 375, 675, 10, and 404, respectively; the numbers of data points for the seqFISH+ SVZ dataset are 9817, 50, 3337, 1424, 1553, 7798, 258, and 291, respectively. **f** Schematic illustration of the procedure and rationale for single-cell ST data to evaluate predicted downstream target and pathways underlying LRIs. **g** Performance comparison of SpaTalk on the inferred downstream targets with other methods (NicheNet, CytoTalk, and CellCall) over the STARmap and seqFISH+ datasets. The *P*-value represents the significance of enriched pathways or biological processes from the KEGG and Reactome databases using inferred downstream targets with the Fisher-exact test. For the boxplots (minima, 25th percentile, median, 75th percentile, and maxima) from SpaTalk to CellCall, the numbers of data points for the STARmap dataset are 25, 1203, 66, and 68, respectively; the numbers of data points for the seqFISH+ OB dataset are 23,132, 19,209, 5168, and 27,668, respectively; the numbers of data points for the seqFISH+ SVZ dataset are 121,693, 25,216, 15,454, and 225,231, respectively.

observed, ranking first for both evaluation indices for STARmap datasets (Fig. 2d). Similarly, SpaTalk obtained a higher median −log₁₀ *P*-value and co-expression percent on the seqFISH+ OB and SVZ datasets but not for SpaOTsc[16], CellPhoneDB[37], and CellChat[38] on individual evaluation indices (Fig. 2e).

Considering the different default LRI database utilized in cell−cell communication inference methods, we benchmarked the performance of SpaTalk and other methods by unifying the LRI database, wherein CellTalkDB[17], CellPhoneDB[37], CytoTalkDB[7], CellChatDB[38], and CellCallDB[8] were used by turns. As shown in Supplementary Fig. 3e, several methods showed decent performance on some individual LRI databases. For example, Giotto obtained the highest median −log₁₀ *P*

among all methods over the STARmap dataset based on CellPhoneDB and CellCallDB, while CytoTalk is the top-ranked method over the STARmap dataset based on CellPhoneDB and CellChatDB considering the median co-expression percent. Over the seqFISH+ OB dataset based on CellChatDB, both of CellPhoneDB and CellChatDB perfectly identified the significantly proximal LR pairs in space. For SpaOTsc, it exhibited the highest median co-expression percent over the seqFISH+ SVZ dataset across all LRI databases. Nevertheless, SpaTalk obtained the most times of the first place across the benchmarked datasets and the underlying LRI databases, outperforming other existing methods for inference of spatially proximal LR pairs that mediating cell−cell communication in space.

Next, we compared the performance of SpaTalk for inference of intracellular signal pathways of the receiver cell type triggered by the LRI with those of NicheNet, CytoTalk, and CellCall that also infer the downstream targets of LRIs. We reasoned that a more accurate method would be more likely to enrich the receptor-related biological processes or pathways using the inferred downstream target genes in the receiver cell type (Fig. 2f); hence, the Fisher-exact test was adopted for pathway enrichment analysis with the KEGG and Reactome databases on target genes in receivers. The target genes inferred by all methods enriched the most intracellular pathways or biological processes triggered by the inter-cellular LRI (Fig. 2g). Nevertheless, SpaTalk exhibited the top-ranked performance over three benchmarked datasets, exceeding other existing methods in inference of the LRT signal network.

In addition, the computation time for each method was also evaluated. For the decomposition step, SpaTalk and other deconvolution six methods were compared over four simulated (STARmap, MERFISH, seqFISH+ OB and SVZ) and one real spot-based (10x Visium) ST datasets, wherein the computation time of SpaTalk was within minutes similar to RCTD and Seurat, outperforming SPOTlight, deconvSeq, Stereoscope, and Cell2location (Supplementary Table 1). For the inferring cell–cell communication step, SpaTalk and other seven methods were benchmarked over three single-cell ST datasets (STARmap, seqFISH+ OB and SVZ) and one reconstructed single-cell ST dataset (10x Visium) with SpaTalk. As shown in Supplementary Table 2, the computation time of SpaTalk, Giotto, SpaOTsc, NicheNet, CellCall, and CellPhoneDB were all within minutes, superior to that of CytoTalk and CellChat. In short, the present results indicated that the SpaTalk is relatively accurate and efficient method for dissecting the cell-type composition of ST data and inferring spatially resolved cell–cell communications.

## Metabolic modulation of periportal hepatocytes on pericentral hepatocytes

We first applied SpaTalk to the Slide-seq ST dataset (v1) of the mouse liver covering 17,545 unique genes among 25,595 spots in space (Fig. 3a). To explore the cell-type composition of Slide-seq data, a mouse liver scRNA-seq reference integrating the non-parenchymal cells from the Mouse Cell Atlas (MCA)[39] and the parenchymal hepatic cells from the "GSE125688 [https://www.ncbi.nlm.nih.gov/geo/query/acc.cgi?acc=GSE125688]" dataset[40] were utilized (Supplementary Fig. 4a), containing 6,029 cells, including major immune cells such as macrophages (Macro), and the pericentral and periportal hepatocytes. The reconstructed single-cell ST atlas was perfectly accordant with the original outcome obtained by Slide-seq (Fig. 3b), wherein the expression of known marker genes[41] and the percent for each cell type were highly correlated across spots (Supplementary Fig. 4b–d), such as the pericentrally and periportally zonated genes Cyp2e1 and Pck1 (Fig. 3c). Immune cells were hardly observed in each spot, with pericentral and periportal hepatocytes accounting for the major proportion across spots (Supplementary Fig. 4e); the same phenomenon was observed in recently published ST data of the healthy liver[42].

The cell–cell communications between pericentral and periportal hepatocytes were further explored by SpaTalk (Fig. 3d). Both hepatocyte types secrete and receive multiple ligands for their communication, forming spatially distributed metabolic cascades to cooperatively optimize the metabolic environment. For instance, with the gradient expression of enzymes in sequential lobule layers, pericentral hepatocytes perform the primary steroid, alcohol, and lipid metabolic processes, while periportal hepatocyte mainly carry out the small-molecule and monosaccharide biosynthetic processes, amino acid and triglyceride metabolic processes, and gluconeogenesis (Fig. 3e, f), in line with the variable functions of zonated hepatocytes residing in the central and portal veins[43]. Notably, the periportal hepatocytes substantially expressed more ligands, including

epidermal growth factor (Egf), transforming growth factor alpha (Tgfa), heparin-binding EGF-like growth factor (Hbegf), insulin-like growth factor 1 (Igf1), and vascular endothelial growth factor A (Vegfa), to promote the growth of pericentral hepatocytes. As the blood flows from the portal vein toward the central vein, this could reflect the fact that periportal hepatocytes respire most of the oxygen, leading to decreased oxygen concentrations along the lobule axis; thus, periportal hepatocytes secrete numerous growth factors via paracrine signaling to modulate the pericentral hepatocytes for the prevention of hypoxia and the maintenance and amelioration of pericentrally metabolic functions[43].

Taking the LRI of Apob-Cd36 as an example, the spatially resolved cell–cell communications between periportal and pericentral hepatocytes mainly occurred across the mid-lobule layers (Fig. 3g). The gene product of Apob is an apolipoprotein of chylomicrons and low-density lipoproteins highly involved with the regulation of lipids and fatty acids metabolism through CD36, indicating the modulation of periportal hepatocytes on the metabolic microenvironment sensed by pericentral hepatocytes. From the reconstructed intracellular signal propagation network triggered by the Apob-Cd36 interaction (Fig. 3h), sequential target TFs were activated, including Ahr that regulates xenobiotic-metabolizing enzymes such as cytochrome P450, and Nr1h4 that regulates the expression of genes involved in bile acid synthesis and transport, in agreement with the corresponding module score of pericentral hepatocytes in space (Fig. 3i). The LRT network was also remarkably enriched in the AMPK and PPAR signaling pathways, which play crucial roles in the regulation of energy and metabolic homeostasis, suggesting the spatially fine-tuned cell–cell communications along the portal-central lobule axis for minimizing risks to pericentral hepatocytes[43].

## Identification of signal transmission among glomerular cells in kidney

SpaTalk was then applied to investigate and visualize the intraglomerular communications over the Slide-seq ST dataset (v2) of the mouse kidney[44] (Fig. 4a), including data of 20,591 sequenced genes for 27,044 spots in space covering the spatial axis of collecting duct intercalated cells (CD-IC), collecting duct principal cells (CD-PC), distal convoluted tubules (DCT), endothelial cells (Endo), fibroblasts (FB), granular cells (GC), macrophages (Macro), mesangial cells (MC), immune cells, proximal convoluted tubules (PCT), podocytes (Pod), thick ascending limb (TAL), and vascular smooth muscle cells (vSMC) from cortex of kidney to renal medulla. As shown in Fig. 4b, SpaTalk identified the spatial signal transmission among Pod, MC, and Endo in glomerulus. For example, the direct cell–cell communication mediated by the pleiotrophin (Ptn) and protein tyrosine phosphatase receptor type B (Ptprb) interaction was observed from MC and Pod to Endo, wherein Ptn is a secreted growth factor that can bind Ptprb, which is known to be involved with adherens junction stimulating endothelial cell migration and maintaining proper glomerular function[45]. Besides, collagen and Notch signaling were also identified forming the matrix that provides structural support for the glomerulus, which are necessary for proper glomerular basement membrane formation and glomerular development[45,46]. Concordantly, these identified LRIs among glomerular cells are associated with multiple biological processes and pathways that play vital roles in the regulation of physiological kidney development and glomerular filtration function in the urinary system[47,48], including tube morphogenesis, positive regulation of cell adhesion, urogenital system development, and morphogenesis of a branching epithelium (Fig. 4c).

Notably, the Pod-Endo communication mediated by Vegfa signaling was significantly enriched in space (Fig. 4b), in accordance with the fact that Vegfa is vital for the formation and maintenance of select microvascular beds within the kidney[49]. It is known that Vegfa, normally produced by healthy podocytes, has been shown to be a critical

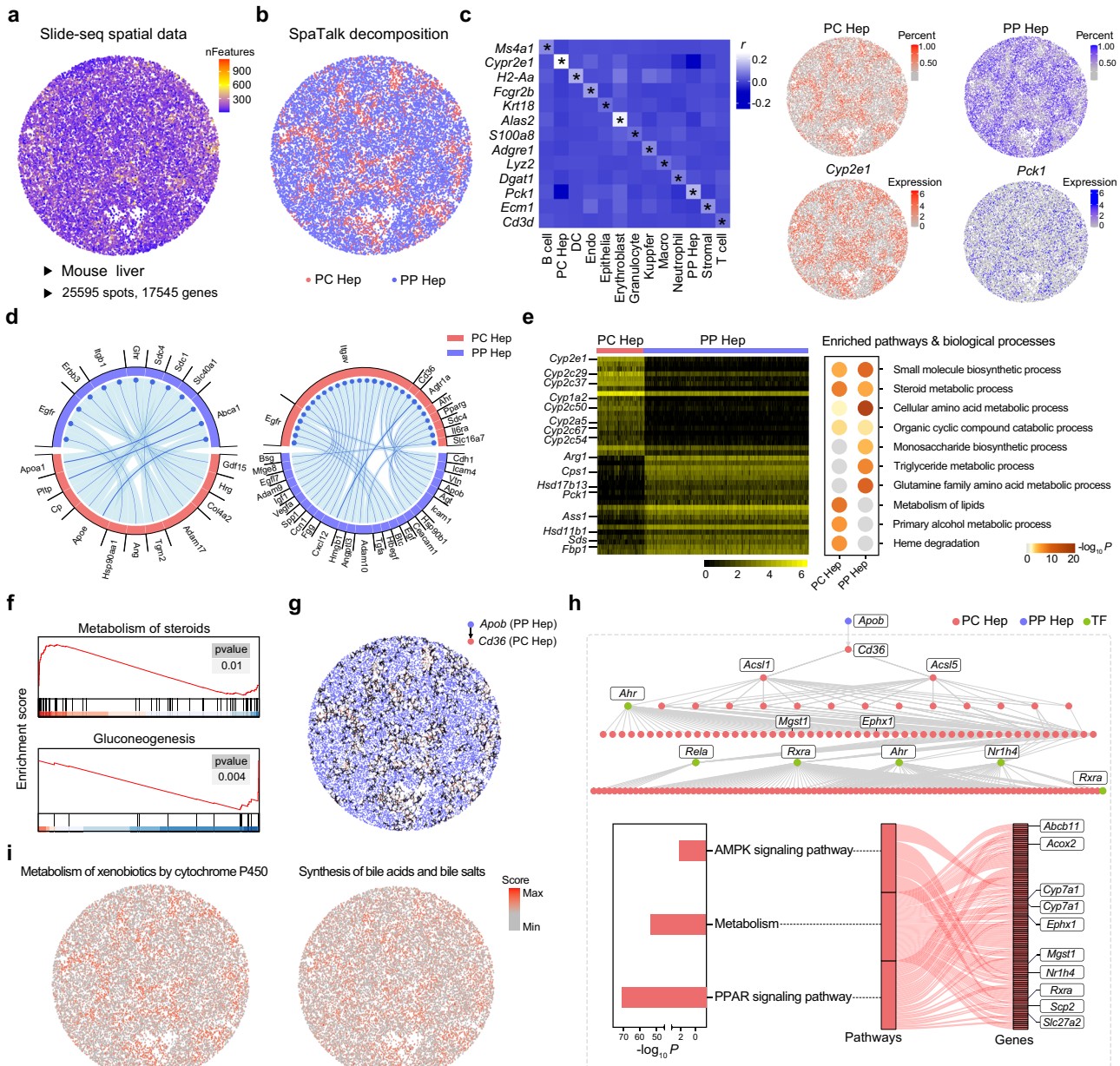

**Fig. 3 | Modulation of periportal hepatocytes on the metabolic micro-environment sensed by pericentral hepatocytes. a** Slide-seq spot-based ST dataset of the mouse liver involving 25,595 spots and 17,545 genes. **b** Cell-type decomposition by SpaTalk. PC, pericentral; PP, periportal; Hep, hepatocytes. **c** Scaled Pearson's correlation coefficients between the expression of known marker genes and the percent for each cell type. Endo, endothelial cells; DC, dendritic cells; Macro, macrophages. **d** Enriched LRIs that mediate cell–cell communications between pericentral and periportal hepatocytes. **e** Significant differentially expressed genes (DEGs) between periportal and pericentral hepatocytes assessed with the Wilcoxon test and the corresponding significantly enriched biological processes and pathways determined with the Metascape web tool. Representative DEGs are labeled beside the heatmap. **f** Significantly activated pathways in pericentral (up) and periportal (down) hepatocytes determined by Gene Set Enrichment Analysis (GSEA). **g** Communications from periportal hepatocytes to pericentral hepatocytes mediated by the *Apob-Cd36* interaction in space. **h** Intracellular signal pathway inferred by SpaTalk over Slide-seq data and the significantly enriched pathways over the ligand–receptor–target network. **i** Inferred module score of each hepatocyte type over the pathway signatures determined with Seurat.

regulator of glomerular development and function and precise expression of the amount of *Vegfa* is required for adequate barrier function[50]. As the kinase insert domain receptor (*Kdr*) of *Vegfa*, *Kdr* functions as the main mediator of VEGF-induced endothelial proliferation, survival, migration, tubular morphogenesis and sprouting. Concordantly, the Pod-Endo communication mediated by *Vegfa-Kdr* was also identified in other mouse kidney ST data (Fig. 4e). In addition, most shared LRIs mediating the intraglomerular communications in space were also observed across slides of human and mouse kidney (Supplementary Fig. 5a), suggesting the robustness and universality of the spatially resolved cell–cell communications inferred by SpaTalk.

We then applied SpaTalk to another spot-based ST dataset of the mouse kidney (10x Visium), reaching up to 19,465 unique genes among 3,124 spots in space (Fig. 4f). Leveraging previously published adult mouse kidney cell taxonomy by single-nucleus RNA-seq data[51] (Supplementary Fig. 5b), SpaTalk reconstructed the spatial transcriptomics atlas at single-cell resolution for Slide-seq data, which showed consistent spatial localization of glomerular and other cells (Fig. 4f and

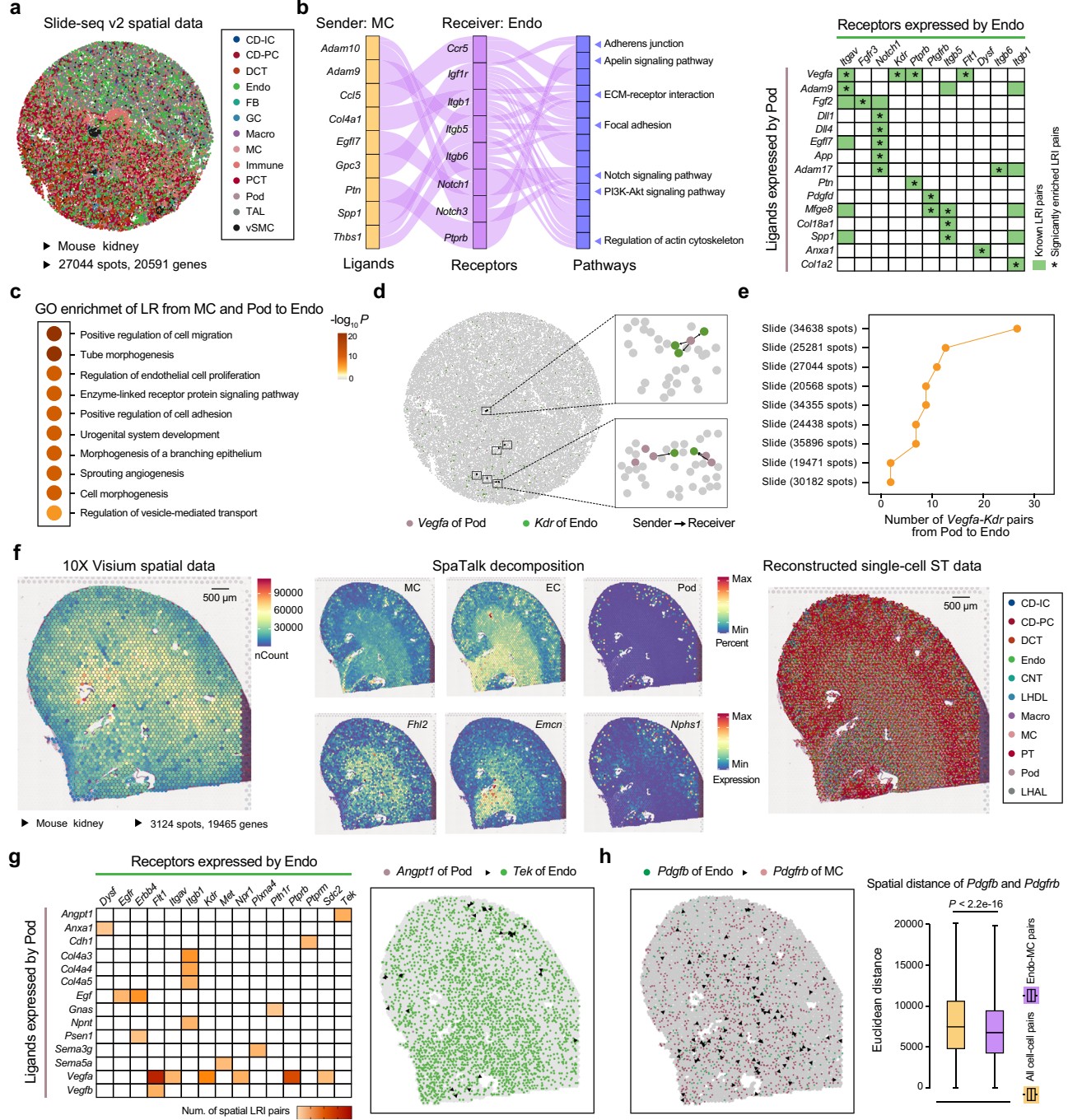

**Fig. 4 | Identification of spatial signal transmission among glomerular cells in kidney. a** Slide-seq (v2) ST dataset of the mouse kidney involving 27044 spots and 20591 genes. CD-IC, collecting duct intercalated cells; CD-PC, collecting duct principal cells; DCT, distal convoluted tubules; Endo, endothelial cells; FB, fibroblasts; GC, granular cells; Macro, macrophages; MC, mesangial cells; PCT, proximal convoluted tubules; Pod, podocytes; TAL, thick ascending limb; vSMC, vascular smooth muscle cells. **b** Significantly enriched LRIs that mediate cell−cell communications among MC, Endo, and Pod inferred by SpaTalk with $P < 0.05$. The $P$-value represents the significance of spatial proximity of LRIs using the permutation test. Top 20 LRI pairs were plotted for Pod-Endo communications. **c** Significantly enriched Gene ontology (GO) biological processes determined with the Metascape web tool for the ligands and receptors from MC and Pod to Endo inferred by SpaTalk. **d** Spatial **d**istribution of the *Vegfa-Kdr* pairs between the Pod senders and Endo receivers. **e** Number of *Vegfa-Kdr* pairs

from Pod to Endo in space across other slides of mouse kidney. **f** Mouse kidney ST dataset generated from 10x Visium involving 3124 spots and 19465 genes and the reconstructed single-cell ST data by SpaTalk. The percent of MC, Endo, and Pod as well as the expression of the corresponding known markers were plotted. PT, proximal tubule; CNT, connecting tubule; LHDL, loop of Henle descending loop; LHAL, loop of Henle ascending loop. **g** Significantly enriched LRIs that mediate Pod-Endo communications. Top 20 LRIs (left) and spatial distribution of the *Angpt1-Tek* pairs between the Pod senders and Endo receivers were plotted. **h** Communications of Endo-MC mediated by the *Pdgfb-Pdgfrb* interaction in space and the spatial distances of *Pdgfb-Pdgfrb* in Endo-MC and all cell−cell pairs, respectively. *P*-value was calculated with the one-sided *t*-test. The numbers of data points (minima, 25th percentile, median, 75th percentile, and maxima) for the boxplots from left to right are 2,237,963 and 405,756, respectively.

Supplementary Fig. 5c, d). Compared to Slide-seq (v2) data, proximal tubules (PT), connecting tubule (CNT), loop of Henle descending loop (LHDL), and loop of Henle ascending loop (LHAL) were also observed in the 10x Visium data. Consistently, most LRIs mediating intraglomerular communications in Slide-seq (v2) data were also found in 10x Visium data (Fig. 4g and Supplementary Fig. 5e). Similar to the *Vegfa* paracrine system, Angiopoietin-1 (*Angpt1*) is expressed by podocytes and its cognate tyrosine kinase receptor (*Tek*) is expressed by the glomerular endothelial cells, which plays an indispensable role in glomerular health and maintenance of the filtration barrier in physiological conditions[47]. Moreover, the direct communication between Endo to MC mediated by platelet-derived growth factor B and its receptor (*Pdgfb-Pdgfrb*) was significantly enriched (Fig. 4h), in accordance with the classical studies that have delineated a key role for the *Pdgfb* in communication between the glomerular endothelium and nearby mesangial cells[45,47].

### Spatial characterization of cell types over 10x Visium data

Given the widely used 10x Visium tool in ST studies, we applied SpaTalk to another human skin squamous cell carcinoma (SCC) ST dataset published by Ji et al.[22], who profiled SCC and matched normal tissues via 10x scRNA-seq and used Visium to identify a tumor-specific keratinocyte (TSK) in the tumor (Fig. 5a). Using the matched SCC scRNA-seq data of patient 2 as reference, the optimal cell-type composition for each spot was deconvoluted by SpaTalk, which exhibited a similar characterization with that histologically assessed from hematoxylin and eosin-stained frozen sections (Fig. 5b). The percent of TSK inferred by our method was compared to the TSK score based on markers (e.g., MMP10, PTHLH, LAMC2, and IL24) defined by Ji et al.[22] across 646 spots (Fig. 5c). A high correlation between the TSK percent and score was observed (Fig. 5d). Moreover, the percent of inferred cell types was prominently associated with the expression of known marker genes, indicating the accuracy of SpaTalk for cell-type decomposition (Supplementary Fig. 6a, b).

Next, we reconstructed the single-cell ST profile by assuming a total of 30 cells in each spot according to a recent review[11], which covered the main epithelial cells, including differentiating, cycling, and basal keratinocytes; melanocytes; fibroblasts (FB); Endo; natural killer (NK) cells; and T cells (Fig. 5e). Despite the asymmetrical distribution for most cell types, TSK, FB, and Endo showed specific patterns of locations in space, which were highly adjacent in some tumor areas (Fig. 5f), forming direct cell–cell communications in the tumor microenvironment (TME). By filtering cells from the TSK leading spots (score ≥ 0.8), we found that TSKs reside within a fibrovascular niche, resulting in high co-localization of TSKs, FB, and Endo at the TSK leading spots (Fig. 5g), in line with the previous findings[22]. Moreover, we used SpaTalk to investigate the cell-type composition over the ST data of another SCC patient. Despite a low percentage across 621 spatial spots, most TSKs centered on a handful of corner spots in space, exhibiting a highly consistent distribution with TSK scores (Fig. 5h and Supplementary Fig. 6c). Unsurprisingly, the fibrovascular niche was also observed in the TSK leading spots of patient 10 with clear spatial co-localization (Supplementary Fig. 6d), indicating the close cell–cell communication among TSKs, FB, and Endo in the TME underlying the occurrence and development of SCC.

### Reconstruction of TSK-stroma communications in space

To dissect the underlying LRI mediating the spatially resolved cell–cell communications between TSKs and stromal cells of the fibrovascular niche in the TME, we applied SpaTalk to infer the communications between TSK-FB and TSK-Endo pairs over the decomposed single-cell ST data of SCC in patient 2, including the top-ranked 20 LRIs based on the integrated inter-cellular and intracellular scores (Fig. 6a). Consistent with a TSK-fibrovascular niche, prominent TSK signaling to FB and Endo was mediated by several common ligand–receptor pairs,

including VEGFA-NPR1, VEGFB-NPR1, PGF-SDC1, and CDH1-ITGAE, associated with tumor angiogenesis. Additionally, TSKs modulate FB through secreting matrix metallopeptidase (MMP)1 and MMP9, which are linked to tumor metastasis via cellular movement and extracellular matrix (ECM) disassembly. Conversely, FB and Endo prominently co-expressed numerous ligands such as MDK, HGF, HMGB1, and THBS1, matching TSK receptors that promote the proliferation and differentiation of TSKs (Fig. 6b and Supplementary Table 3). Further supporting TSKs as an epithelial mesenchymal transition (EMT)-like population, SpaTalk predicted that the widely expressed TGFB1 regulates TSKs. TSK receptors corresponding to additional ligands from FB and Endo included several integrins (e.g., ITGA5 and ITGB6) and nectins (e.g., NECTIN1 and NECTIN2), highlighting other pathways associated with EMT and epithelial tumor invasion[52,53]. Notably, similar LRIs mediating the TSK-stroma communications in space were also observed in another SCC patient (Supplementary Fig. 7a).

Next, we focused on a region of interest (ROI) covering 42 spatial spots in space for in-depth exploration of TSK-stroma communications, which exhibited a high total score of TSK, FB, and Endo according to their signature genes, occupying the major part in the ROI (Fig. 6c). By mapping cancer-associated fibroblast (CAF) markers to the cells in space, the majority of FB in the ROI highly expressed the known CAF marker genes (e.g., VIM, FAP, POSTIN, and SPARC) (Fig. 6d), hinting at the transformation of FB to CAFs induced by the adjacent TSKs and conversely supporting the stemness of TSKs via direct cell–cell communication in space (Supplementary Fig. 7b). Notably, the TSKs appear to be heterogeneous with respect to the broad range of EMT scores in the ROI; thus, TSKs were further classified into 268 EMT-like and 268 EMT-unlike populations (Supplementary Fig. 7c). By comparing their differentially expressed genes, EMT-like TSKs were dramatically enriched with ECM organization, proteoglycans in cancer, regulation of cell adhesion, and the VEGFA-VEGFR2 signaling pathway, representing more invasive properties compared with EMT-unlike TSKs (Fig. 6e).

Additionally, EMT-like TSKs appear to be more communicative with surrounding CAFs in the TME in light of the greater number of cell–cell pairs over the LRIs that prominently mediate TSK-stroma communications in space, such as LAMB3-ITGB1, LAMA3-ITGB1, LAMC2-ITGB1, and MMP1-CD44 (Fig. 6f and Supplementary Fig. 7d–f). Metastasis-related laminins are essential for formation and function of the basement membrane[54], whereas MMPs are involved in ECM breakdown, both contributing to the aggravated malignancy of tumors. Moreover, CD44 expression on CAFs plays a supporting role in the induction of cellular stemness[55], wherein CAFs have a preference of cell–cell communications with EMT-like TSKs in space (Fig. 6g). Interestingly, EMT-unlike TSKs notably exhibited more communicative cell–cell pairs with Endo, whereas EMT-like TSKs exhibited significantly more communicative cell–cell pairs with CAFs over the matched LRIs (Fig. 6h), consistent with the observed contribution of CAFs to EMT in a broad range of tumors[56,57].

## Discussion

We have demonstrated the capabilities of SpaTalk to infer and visualize spatially resolved cell–cell communications mediated by significantly enriched LRIs under normal and disease states over existing representative datasets, including the single-cell ST data generated from STARmap, MERFISH, and seqFISH + , and the spot-based ST data obtained via Slide-seq and 10x Visium.

There are two principles to decode the mechanisms of cell–cell communications: ligand–receptor proximity and LRT co-expression[11]. In a given tissue niche, cells are more likely to communicate with each other when they are spatially adjacent and activate downstream target genes in the receiving cell triggered by the LRI in proximal cells; thus, ST data are well suited to apply the two principles for inferring intercellular communications. Accordingly, our proposed SpaTalk realizes

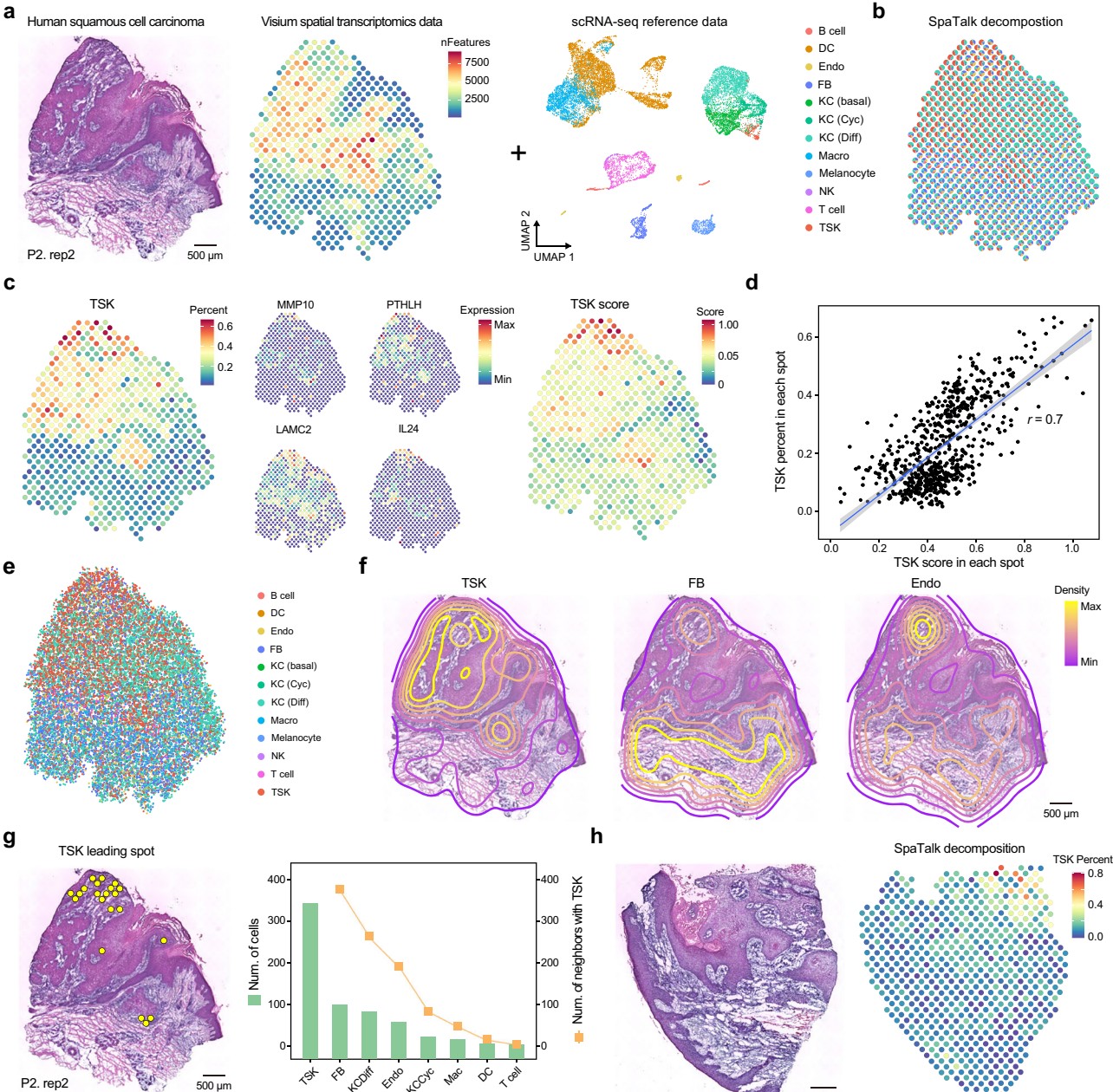

**Fig. 5 | Spatial characterization of tumor and stromal cells in human squamous cell carcinoma Visium data with SpaTalk. a** Visium spot-based ST dataset of human skin SCC in patient 2 with the matched scRNA-seq dataset involving the main keratinocytes (KC), stromal cells, and immune cells. **b** Cell-type decomposition by SpaTalk. Cyc, cycling; Diff, differentiating; NK, natural killer; FB, fibroblasts; TSK, tumor-specific keratinocytes. **c** TSK percent and TSK score across spatial spots. The expression of known TSK markers is plotted. **d** Pearson's correlation coefficient between the TSK percent and TSK score. The gray band represents the 95% confidence interval of the mean value for the fitting straight line. **e** Cell-type decomposition by SpaTalk at single-cell resolution for the spot-based human skin SCC ST data. **f** Contour plot of TSK, FB, and Endo based on the reconstructed single-cell ST atlas by SpaTalk. **g** TSK leading spots with a TSK score >0.8 in space. The bar chart represents the number of different cell types and the line chart represents the number of neighbors adjacent to TSKs among the TSK leading spots. **h** Visium spot-based ST dataset of human skin SCC in patient 10 and the cell-type decomposition by SpaTalk showing the percent of TSKs across 621 spatial spots.

the integration of these two principles by incorporating the KNN and cell–cell graph network to filter spatially proximal cell pairs and corresponding LRIs, followed by utilizing the knowledge graph algorithm to model the LRT signal propagation process. Consequently, the performance of SpaTalk was superior to that of other methods over the benchmarked ST datasets with respect to several evaluation indices, demonstrating the reliability of the two principles in decoding cellular cross-talk, especially for juxtracrine and paracrine communication.

Importantly, SpaTalk is applicable to either single-cell or spot-based ST datasets generated from mainstream ST technologies. For the former, SpaTalk assigns a label to each cell by selecting similar cell types with the top-ranked weight via NNLM for single-cell ST data, generating the ST atlas at single-cell resolution with known cell types for the subsequent inference of cell–cell communications. For spot-based ST data, SpaTalk selects and maps the optimal combination of cells in accordance with the decomposed optimal weight/percent of

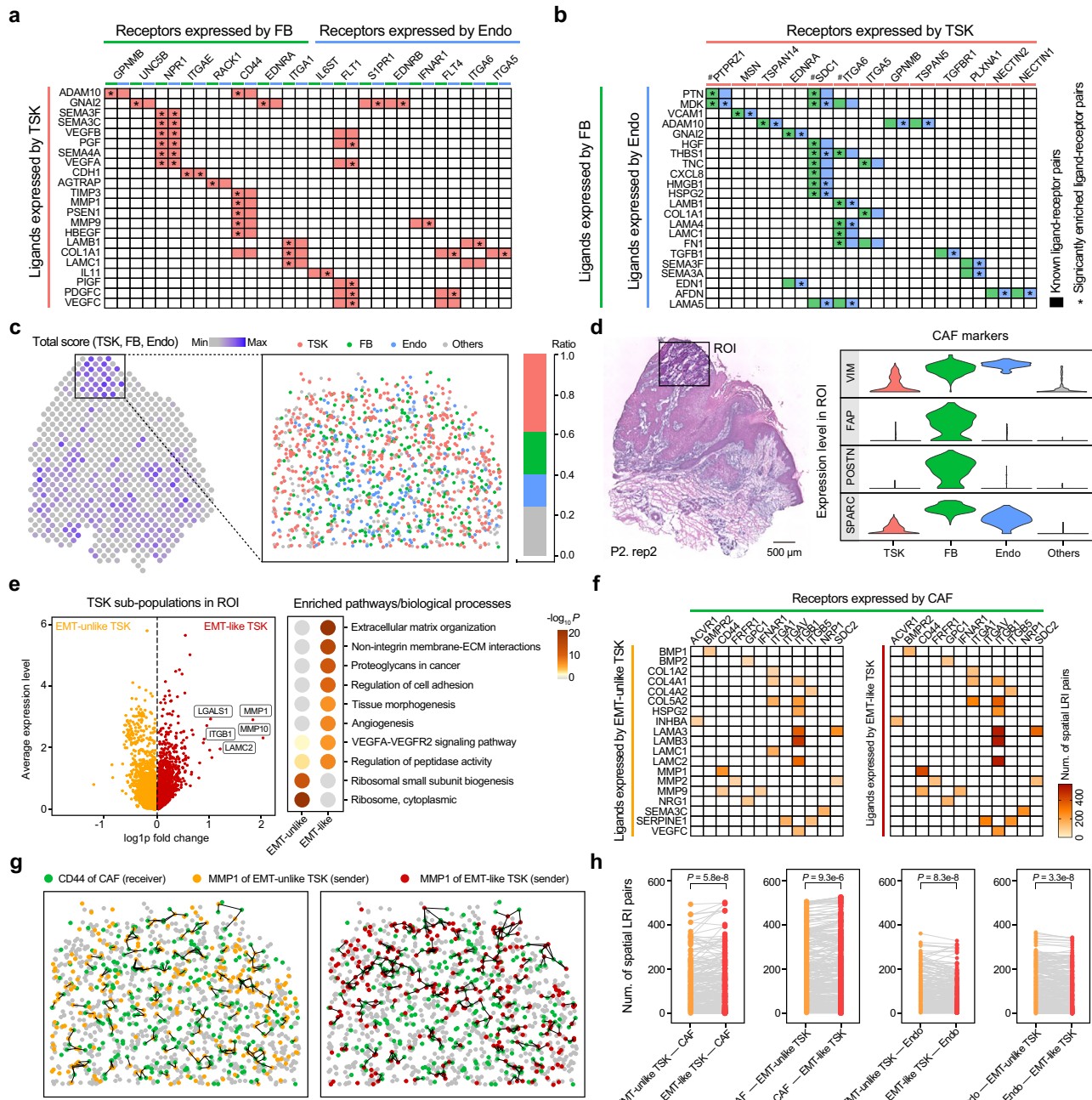

**Fig. 6 | Reconstruction of cell-cell communications between TSK subpopulations and stromal cells in space with SpaTalk. a** Top 20 inferred LRIs that mediate cell-cell communication from the TSK senders to the FB and Endo receivers. Colored blocks represent the known ligand-receptor pairs in CellTalkDB. The asterisk represents the significantly enriched LRIs determined by SpaTalk. **b** Top 20 inferred LRIs that mediate cell-cell communication from the FB and Endo senders to TSK receivers. **c** Region of interest (ROI) covering 42 spatial spots in space with a high total score of TSK, FB, and Endo according to their signature genes. The point plot shows the decomposed single-cell ST atlas determined by SpaTalk. The number sign represents the receptor contributing to the proliferation and differentiation of TSK. **d** Expression of known cancer-associated fibroblast (CAF) markers across TSKs, FB, Endo, and other cells. **e** Differentially expressed genes (DEGs) between EMT-like and EMT-unlike TSKs and the corresponding enriched biological processes and pathways. Representative DEGs are labeled beside the point. **f** Number of cell-cell pairs over the LRIs from the EMT-like and EMT-unlike TSKs to CAFs. **g** Communications from EMT-like and EMT-unlike TSKs to CAFs mediated by the MMP1-CD44 interaction in space. **h** Comparison of communications among CAFs, Endo, EMT-like TSKs, and EMT-unlike TSKs with respect to the number of cell-cell pairs in space over the matched LRIs evaluated with a paired two-sided Wilcoxon test.

cell types via NNLM and the transcriptome profiles of spatial spots to reconstruct the ST atlas at single-cell resolution with known cell types. Notably, applications of SpaTalk to the mouse cortex and liver datasets sequenced by STARmap and Slide-seq, respectively, revealed the evidential LRIs in space that mediate the spatially resolved cell-cell communications contributing to normal physiological processes. Moreover, exploration of SpaTalk on the human skin SCC dataset

obtained from 10x Visium identified the variable preference of communication among tumor subpopulations, CAF, and Endo. These cases convincingly demonstrate the universality of SpaTalk in decoding the mechanism of cell-cell communications in space underlying normal and disease tissues for single-cell and spot-based ST data.

As unmatched scRNA-seq and ST data would directly influence the cell-type decomposition, an important feature of SpaTalk is the ability

to assign a spot/cell into an unsure category considering the unseen cell types in the scRNA-seq reference. For example, the scRNA-seq reference for the Slide-seq mouse cortex ST data was obtained from another repository for the same tissue, resulting in numerous unsure cells by assuming one cell in each spot in terms of the high resolution (10 μm) of Slide-seq technology that almost approaches single-cell resolution. Additionally, the low gene coverage of several spatial spots severely affects the regression model, which were regarded as the unsure type by SpaTalk. However, with respect to the human skin SCC datasets, SpaTalk removed the unsure type for the matched scRNA-seq and ST data. As the matched multi-modal datasets will undoubtedly become greater in number, application of SpaTalk and similar methods will be required for accurate inference of spatially resolved cell–cell communications.

Additionally, SpaTalk characterizes the spatial distribution for each cell type within the reconstructed ST at single-cell resolution through the contour plot of cellular density in space, which enables analyzing the proximal relationship between paired cell types. Moreover, SpaTalk enables the statistical analyses and visualization of spatially proximal LRIs in space, forming a dynamic cell–cell communication network. Currently, it is hard to analyze and visualize the LRI at single-cell resolution for scRNA-seq data, wherein the common practice is to interpret the LRI for paired cell types. By incorporating spatial information, SpaTalk displays the enriched LRI at single-cell resolution via the spatially proximal co-expressed cell pairs, offering an informative approach for the analysis and visualization of the LRI and its mediated cell–cell communication underlying the disease pathology from a different perspective, as shown in the application of SpaTalk to the human skin SCC datasets.

Adding spatial constraints in cell–cell communication inference is critical to the spatial analysis of juxtracrine and paracrine communications. However, this constraint inevitably causes the failure of inferring long-range communications such as endocrine and telecrine signaling. Classification of LRIs into short-range and long-range communications with prior knowledge might be helpful to infer the comprehensive communication categories computationally. Moreover, it is potentially beneficial to include other omics data with the increasing multi-modal datasets generated from state-of-the-art technologies such as 10x Multiome and Digital Spatial Profiling[58] in studying spatially regulated cell–cell communications. Thus, more reliable computational models might be needed for more accurate integration of multi-modal data and inference.

## Methods

### Datasets
For STARmap[19], the single-cell ST data of the mouse cortex (20180410-BY3_1kgenes) was obtained from a "public data portal [https://www.dropbox.com/sh/f7ebheru1lbz91s/AABYSSjSTppBmVmWl2H4s_K-a?dl=0]". For MERFISH[29], the single-cell ST data of the naïve female mouse (Animal_ID: 1, Bregma: 0.26) hypothalamic preoptic region was downloaded from "Dryad [https://datadryad.org/stash/dataset/doi:10.5061/dryad.8t8s248]". For seqFISH+[30], the single-cell ST data of the mouse cortex and olfactory bulb were retrieved from the "Github repository [https://github.com/CaiGroup/seqFISH-PLUS]". For Slide-seq[20,21], the spot-based ST data of the mouse liver (Puck_180803_8) and somatosensory cortex (Puck_200306_03) were obtained from the Broad Institute Single Cell Portal "SCP354 [https://singlecell.broadinstitute.org/single_cell/study/SCP354/slide-seq-study]" and "SCP815 [https://singlecell.broadinstitute.org/single_cell/study/SCP815/highly-sensitive-spatial-transcriptomics-at-near-cellular-resolution-with-slide-seqv2]", respectively, and the human and mouse kidney ST datasets[44] were collected "here [https://cellxgene.cziscience.com/collections/8e880741-bf9a-4c8e-9227-934204631d2a]". For 10x Visium, the ST data and scRNA-seq data of human SCC were downloaded from the Gene Expression

Omnibus (GEO) repository: "GSE144240"[22], and the mouse kidney ST and single-nucleus RNA-sequencing data were obtained from 10x Visium Spatial Gene Expression of "'Adult Mouse Kidney (FFPE)' [https://www.10xgenomics.com/resources/datasets]" and "GSE119531"[51], respectively. The mouse liver scRNA-seq data of non-parenchymal and parenchymal hepatic cells were refined from the "MCA [https://figshare.com/articles/MCA_DGE_Data/5435866]"[39] and "GSE125688"[40], respectively. The mouse cortex scRNA-seq data were obtained from "GSE71585"[59].

### Data processing
For the liver scRNA-seq datasets, the non-parenchymal cells and parenchymal hepatic cells were collected from the MCA[39] and "GSE125688"[40], respectively, wherein hepatocytes were classified into pericentral and periportal hepatocytes with principal component analyses and clustering analysis. For the MERFISH dataset, ependymal cells were excluded due to the limited cell number in the section (<2). For human and mouse kidney ST data sequenced by Slide-seq (v2), datasets with at least 15 mesangial cells were included. For other datasets, all cells were included in the filtered matrices. Human and mouse gene symbols were revised in accordance with "NCBI gene data [https://www.ncbi.nlm.nih.gov/gene/]" updated on June 30, 2021, wherein unmatched genes and duplicated genes were removed. For all ST and scRNA-seq datasets, the raw counts were normalized via the global-scaling normalization method LogNormalize in preparation for running the subsequent scDeepSort pipeline.

### SpaTalk algorithm
The SpaTalk model consists of two components: cell-type decomposition and spatial LRI enrichment. The first component is to infer cell-type composition for single-cell or spot-based ST data, and the second component is to infer spatially proximal ligand–receptor interactions that mediate cell–cell communications in space.

**Cell-type decomposition.** To dissect the cell-type composition for the ST data matrix $T$ [$n \times s$] ($n$ genes and $s$ spots/cells), NNLM was first applied to obtain the optimal proportion of cell types using the scRNA-seq data matrix $S[n \times c]$ ($n$ genes and $c$ cells) as the reference with $k$ cell types. Let $Y = \{y_1, y_2, \ldots, y_n\}$ be the expression profile for each spot/cell to establish the following linear model:

$$Y = X\beta + \varepsilon \qquad (1)$$

where $X = [n \times k]$ is the average expression profile generated from $S$ and $\varepsilon$ represents random error. Mean relative entropy loss was then used to measure the difference between the predicted and observed values. Therefore, the objective function can be written as:

$$\arg\min\{\beta \geq 0\}L(Y - X\beta) + \lambda_1 R_1(\beta) + \lambda_\alpha R_\alpha(\beta) + \lambda_2 R_2(\beta) \qquad (2)$$

where $R_1$, $R_\alpha$, and $R_2$ represent the L1, angle, and L2 regularization with non-negative $\lambda_1$, $\lambda_\alpha$, and $\lambda_2$ initialized by zero[23,24]. The model was trained with the above objective function using Lee's multiplicative iteration algorithm[25] with default hyperparameters until convergence or after 10,000 iterations to generate the coefficient matrix $C[k \times s]$.

For single-cell ST data, the cell type with the maximum coefficient was assigned to each cell. For spot-based ST data, let $M$ be the maximum cell number for each spot, which was set to 30 for 10x Visium data and was set to 1 for Slide-seq data according to a recent review[12]. In practice, the optimal cellular combination $\omega$ for each spot was

determined by the following function:

$$\omega_i (i \in \{1, 2, \ldots, k\}) = \begin{cases} [M\beta_i] + 1 & (\{M\beta_i\} \geq 0.5) \\ [M\beta_i] & (\{M\beta_i\} < 0.5) \end{cases} \quad (3)$$

wherein $[M\beta_i]$ and $\{M\beta_i\}$ represent the integer and fractional parts of $M\beta_i$, respectively. For each spot, we randomly selected $m$ ($m = \sum_{i=1}^{k} \omega_i$) cells from $S$ to compare their merged expression profile $\epsilon$ with the ground truth according to the following function:

$$\arg\min\{m \leq M\} \sum_{i=1}^{n} (Y_i - \sum_{j=1}^{m} \epsilon_i^j)^2 \quad (4)$$

To assign a coordinate $(\hat{x}, \hat{y})$ for each sampled cell, we proposed a probabilistic distribution for a given cell in each spot $(x_0, y_0)$ by considering the ratio $(R)$ of the same cell type in $Q$ neighbor spots as the probability to locate the cell into the space with the following function:

$$\hat{x} = x_0 + \alpha d_{min} \cos(\theta\pi/180)/2$$
$$\hat{y} = y_0 + \alpha d_{min} \sin(\theta\pi/180)/2 \quad (5)$$

where $d_{min}$ represents the spatial distance of the closest neighbor spot, and $\alpha \in (0, 1]$ and $\theta \in (0, 360]$ mean the weight for $d_{min}$ and the angle towards the spot center $(x_0, y_0)$, respectively. In detail, the $\theta$ were first determined by the following probability equation:

$$\hat{P}(\theta) = \frac{R_q + 1}{\sum_{i=1}^{Q} (R_i + 1)}, \theta \in (90q - 90, 90q] \quad (6)$$

where $q$ represents the $q$th neighbor spot in $Q$, which was set to 4 in practice denoting that the space centered the spot were split into four areas and the nearest neighbor in each area was filtered. After determining the $\theta$, the corresponding neighbor spot $(x_\theta, y_\theta)$ was selected to determine the probabilistic distribution of $\alpha$ by the following equation:

$$\hat{P}(\alpha) = \begin{cases} (R_{x_0,y_0} + 1)/(R_{x_0,y_0} + R_{x_\theta,y_\theta} + 2), \alpha \in (0, 0.5] \\ (R_{x_\theta,y_\theta} + 1)/(R_{x_0,y_0} + R_{x_\theta,y_\theta} + 2), \alpha \in (0.5, 1] \end{cases} \quad (7)$$

where $R_{x_0,y_0}$ and $R_{x_\theta,y_\theta}$ represent the ratio of the given cell type in each spot and its neighbor spot. By integrating the optimal cellular combinations for all spots, ST data at single-cell resolution were reconstructed for the spot-based ST data.

**Spatial LRI enrichment.** To generate the cell–cell distance matrix $D$, the Euclidean distance between cells was calculated using the single-cell spatial coordinates of ST data. Inspired by Giotto, the KNN algorithm was then applied to each cell to select the $K$ nearest cells from $D$ to construct the cell graph network, wherein $K$ was set to 10 by default. For a given ligand $i$ of the sender (cell type $A$) and a given receptor $j$ of the receiver (cell type $B$), the number of ligand–receptor co-expressed cell–cell pairs ($C_{Ai,Bj}^0$) was obtained from the graph network by counting the 1-hop neighbor nodes of receivers for each sender, resulting in the different number of cell–cell pairs for a given LRI pair between the sender cell type $A$ and the receiver cell type $B$. The permutation test was then performed by randomly shuffling cell labels to recalculate the number of LRI pairs. By repeating this step $Z$ times, a background distribution $C = \{C_{Ai,Bj}^1, C_{Ai,Bj}^2, \ldots, C_{Ai,Bj}^Z\}$ was obtained for comparison with the real interacting score, and the $P$-value was calculated as follows:

$$P_{Ai,Bj} = \mathrm{crad}\{x \in C | x \geq C_{Ai,Bj}^0\}/Z \quad (8)$$

where $P_{Ai,Bj}$ values less than 0.05 were filtered to calculate the intercellular score of LRI from senders to receivers ($S_{Ai,Bj}^{inter} = 1 - P_{Ai,Bj}$). To further enrich the LRIs that activate downstream TFs, target genes, and the related pathways of receivers, the knowledge graph was introduced to model the intracellular signal propagation process. In practice, LRIs from CellTalkDB, pathways from KEGG and Reactome, and TFs from AnimalTFDB were integrated to construct the ligand–receptor–TF knowledge graph (LRT-KG), wherein the weight between entities represents the co-expressed coefficient. Taking the receptor as the query node, we incorporated the random-walk algorithm into the LRT-KG to filter and score the downstream activated $t$ TFs with no more than 10 steps and $Z$ iterations; thus, the probability $p$ for each TF can be calculated with the ratio of successful hits from the query node to the target TF during the $Z$ random walks. By integrating the co-expressed TFs and the corresponding target genes from the LRT-KG, the intracellular score of LRI from senders to receivers can be written as:

$$S_{Ai,Bj}^{intra} = \sum_{k=1}^{t} \theta_k \times p_k / \eta_k \quad (9)$$

where $\theta$ represents the number of targeted genes, $\eta$ represents the step from the receptor to the TF in the LRT-KG. By the sigmoid transformation for $S_{Ai,Bj}^{intra}$, the final score of the LRI from cell type $A$ to cell type $B$ can be written as:

$$S_{Ai,Bj} = \sqrt{S_{Ai,Bj}^{inter} \times S_{Ai,Bj}^{intra}} \quad (10)$$

## Comparison with other methods

STARmap, MERFISH, and seqFISH+ ST data were used to compare the performance of SpaTalk with other existing cell-type decomposition methods. For these single-cell ST data, all cells were split according to the fixed spatial distance and then merged into simulated spots as the benchmark datasets. RCTD[31], Seurat[32], SPOTlight[33], deconvSeq[34], Stereoscope[35], and Cell2location[36] were benchmarked with the default parameters and evaluated with Pearson's correlation coefficient and RMSE over the predicted and real cell-type composition for each spot.

Considering the limited genes of MERFISH ST data, STARmap and seqFISH+ single-cell ST data (including 1020 and 10,000 genes, respectively) were used as the benchmark datasets to compare the performance of SpaTalk with other cell–cell communication inference methods (Giotto[15], SpaOTsc[16], NicheNet[6], CytoTalk[7], CellChat[38], CellPhoneDB[37], and CellCall[8]). The one-sided Wilcoxon test was performed to evaluate the spatial proximity significance of the inferred LRIs by comparing the number of expressed LRIs between sender-receiver pairs and all cell–cell pairs, and the co-expressed percent of the LRI was calculated to evaluate the co-expression level by counting the number of expressed LRIs from senders to receivers from the cell–cell graph network. All methods were benchmarked with the default parameters and default LRI database. All inferred LRIs were unbiasedly evaluated with the above criteria except for the LRIs from SpaOTsc since the number of inferred LRIs was much larger than that of the other methods; thus, the top 1000 LRIs for each cell–cell communication were selected from SpaOTsc according to the final score. In addition, we benchmarked the performance of SpaTalk and other methods by unifying the LRI database, wherein the LRI pairs in CellTalkDB, CellPhoneDB, CytoTalkDB, CellChatDB, and CellCallDB were used by turns with the default parameters.

Given the significantly enriched biological processes or pathways in the receiver cell type, the Fisher's exact test was adopted for pathway enrichment analysis with the KEGG and Reactome databases on the activated genes in receivers using the following function:

$$P = \binom{a+b}{a}\binom{c+d}{c} / \binom{n}{a+c} \quad (11)$$

| | Interested genes | Uninterested genes |
|---|---|---|
| Genes matching the pathway | a | b |
| Genes unmatching the pathway | c | d |

where $n = a + b + c + d$; $a$ is the number of inferred target genes (interested genes) that match the given pathway; $b$ is the number of a given pathway's genes that exclude $a$, namely the uninterested genes that match the given pathway; $c$ is the number of inferred target genes (interested genes) that unmatch the given pathway; $d$ is the number of all genes excluding $a$, $b$, and $c$, namely the uninterested genes that unmatch the given pathway. NicheNet, CytoTalk, and CellCall were benchmarked with the default parameters and the inferred target genes for each LRI were evaluated according to the significance of pathway enrichment.

## Pathway and biological process enrichment

The "Metascape web tool [https://metascape.org/]"[60] was used to perform the enrichment analysis of pathways and biological processes, wherein the top 100 highly expressed genes were selected according to the fold change of the average gene expression. Gene Set Enrichment Analysis (GSEA)[61] was performed using the ranked gene list with the clusterprofiler tool to enrich the significantly activated pathways and biological processes, whose signatures were obtained from the Molecular Signatures Database v7.4 ("MSigDB [http://www.gsea-msigdb.org/gsea/msigdb]"[62], including the gene sets from Gene Ontology (GO) and the canonical pathway gene sets derived from the KEGG, Reactome, and WikiPathways pathway databases.

## Module scoring of hallmarks and signatures

Hallmark scoring of metabolism of xenobiotics by cytochrome P450, synthesis of bile acids and bile salts, TSK, and EMT was performed using the "AddModuleScore" function in Seurat with default parameters. Hallmark pathways and EMT were obtained from MSigDB[62], and the signature genes of the TSK were download from the original publication by Jin et al.[22].

## Statistics

R (version 4.1.1), GraphPad Prism 8, and Python 3.9 were used for all statistical analyses.

## Reporting summary

Further information on research design is available in the Nature Research Reporting Summary linked to this article.

## Data availability

The original data used in this paper can be accessed through the following links: (1) STARmap spatial data of the "mouse cortex [https://www.dropbox.com/sh/f7ebheru1lbz91s/AABYSSjSTppBmVmWl2H4s_K-a?dl=0]";[19] (2) MERFISH data of the "naïve female mouse hypothalamic preoptic region [https://datadryad.org/stash/dataset/doi:10.5061/dryad.8t8s248]";[29] (3) seqFISH+ data of the "mouse cortex and olfactory bulb [https://github.com/CaiGroup/seqFISH-PLUS]"[30] downloaded from Github repository; (4) Slide-seq data of the "mouse liver [https://singlecell.broadinstitute.org/single_cell/study/SCP354/slide-seq-study]"[20], "somatosensory cortex [https://singlecell.broadinstitute.org/single_cell/study/SCP815/highly-sensitive-spatial-transcriptomics-at-near-cellular-resolution-with-slide-seqv2]"[21], and the "human and mouse kidney [https://cellxgene.cziscience.com/collections/8e880741-bf9a-4c8e-9227-934204631d2a]";[44] (5) spatial data and scRNA-seq data of human SCC: GEO accession: "GSE144240";[22] (6) spatial data of the "mouse kidney [https://www.10xgenomics.com/resources/datasets]" downloaded from 10x Visium Spatial Gene Expression; (7) Single-nucleus RNA-sequencing data of the mouse kidney: GEO accession: "GSE119531";[51] (8) mouse liver scRNA-seq data of "non-parenchymal cells [https://figshare.com/articles/MCA_DGE_Data/5435866]"[39] and parenchymal hepatic cells: GEO accession: "GSE125688";[40] (9) mouse cortex scRNA-seq data: GEO accession: "GSE71585"[59]. Molecular Signatures Database was

downloaded from "MSigDB v7.4 [http://www.gsea-msigdb.org/gsea/msigdb]". All other relevant data supporting the key findings of this study are available within the article and its Supplementary Information files or from the corresponding author upon reasonable request. Source data are provided with this paper.

## Code availability

Source codes for the SpaTalk R package and the related scripts are available at "SpaTalk Github [https://github.com/ZJUFanLab/SpaTalk]"[63].

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

## Acknowledgements

This work was supported by the National Natural Science Foundation of China (81973701, X.F.), the Key Program, National Natural Science Foundation of China (81930016, X.X.), the Key Research and Development Program of China (2021YFA1100500, X.X.), the Major Research Plan of the National Natural Science Foundation of China (No.92159202, X.X.), the Natural Science Foundation of Zhejiang Province (LZ20H290002, X.F.), the China Postdoctoral Science Foundation (2021M702828, X.S.) and Alibaba Cloud. The authors gratefully thank Giotto and NicheNet for their inspirations in the development of this project and thank Eric Xihui Lin for his implementation of NNLM package.

## Author contributions

X.F., X.X., and H.C. conceived and designed the study. C.L., X.L., J.L., J.Q., and K.W. collected and analyzed the scRNA-seq and ST data. X.S., H.Y., J.C., and P.Y. implemented the algorithm of Spa-Talk. X.S., C.L., and H.Y. developed the package of SpaTalk. All authors wrote the manuscript, read, and approved the final manuscript.

## Competing interests

The authors declare no competing interests.
