## [Peer Review File · Nature Communications]

REVIEWER COMMENTS

Reviewer #1 (Remarks to the Author):

In this manuscript, the authors present a novel method for inference of cell-cell communication networks from spatial transcriptomics data. The method considers both the spatial proximity and activated downstream signalling in the target cell for inference of relevant ligand-receptor interactions. The spatial proximity is implemented using a knn graph and permutations to calculate significant cell proximity enrichment, in a similar way as other methods such as Giotto, however the authors in addition consider ligand-receptor-target co-expression to score the ligand-receptor interactions that activate downstream signalling, in a similar way as NicheNet. The method is especially relevant for spot-based spatial data where the authors combine deconvolution of cell types in each spot and then inference of cell-cell communication, however some details of the method were not clear to me with regards to spot-based data. In addition, some improvements and additions would make the method more usable and some issues with the comparison with other similar methods should be addressed.

Major comments:

- I was a bit confused with the description about the KNN distance graph. Since the interaction score here is calculated only based on the distance between two cell types from what I understood, this would be the same score for all ligand-receptor pairs between two cell types (sender and receiver). It could be better explained that the inter-cellular score is only calculated based on distance and is not per ligand-receptor pair, but per sender-receiver cell type pair. This is very similar to what is implemented in Giotto and should be acknowledged.
- It was not quite clear to me how spatial coordinates are assigned to each cell type within a spot, could you please explain this part in more details? What does this mean for the knn graph, I saw that you use $K=10$ in your code, if you consider 30 cells in each spot, does that mean that some of them are directly connected between each other in the knn graph and other not, and how are they connected with cells from other spots? Related to this, how is the downstream signalling implemented for spot-based data where there are multiple different cell types within a spot, which would lead to multiple signalling pathways and targets activated for each of those different cell types? How do the results compare when used on scRNA-seq data instead of spatial data without considering spatial proximity?
- It would increase the usability of the method if there is an option for the user to use a method for deconvolution of ST data of their choice and then only use the second component for inference of cell-cell communication.
- Is there an option to change the default number of expected cells in each spot from 30 to another number? The user should be able to change this depending on previous knowledge or whether they can use segmentation to estimate the number.
- Was the same LR database (CytoTalkDB) used for comparison of the performance of different cell-cell communication platforms? I could not find this information in the Methods. Otherwise, it would be difficult to compare the results.
- Cell2location was not used in the benchmarking for the deconvolution step.

Reviewer #2 (Remarks to the Author):

In this manuscript, Shao, Li and Yang et al. introduced a method (SpaTalk) to use single-cell RNA-seq

references to decompose spatial transcriptomic data and infer short-range cell-cell signalling and their downstream regulatory networks. They also benchmarked their method against existing tools with similar goals using published data.

Major concerns:

1. In Figure 3, the authors used STARmap brain data to infer short-range signaling between neurons and other cells based on the locations of cell bodies captured by STARmap. However, due to the shape of neurons, interactions can usually happen far away from the cell bodies through axons and synapses that current segmentation methods cannot well capture. The authors should address that point or may consider focusing on other tissues.
2. The authors compared their method against multiple decomposition and cell-cell interaction algorithms. However, cell2location (decomposition), cellphoneDB (interaction) and cellchat(interaction) have been missed. These packages have been widely used. It would be more convincing to benchmark against them, especially cell2location.
3. The authors should also compare computational time between SpaTalk and other methods.
4. For spot-based spatial data, the authors introduced arbitrarily generated locations of single-cells (x,y) around each spot (x0,y0) (Figure 6c, Line 598-600) which increases the noise of the data. Is it a must? Could neighbor spots around each spot also be used to help decide alpha and theta?
5. Handling biological replicates is a key challenge for signaling inference that spatial algorithms start to meet (One example is the multinichenetr algorithm being developed <https://github.com/saeyslab/multinichenetr>). How would the cell decomposition step and the graph learning step handle replicates and how robust will it be?

Minor:

1. Fig 1c: "siganling" should be "signaling"
2. For visualizing cell-cell crosstalks, the arrows on top of cells are very difficult to see (such as Fig 4g, 6g, and supplementary Fig 6). The authors can consider just using lines without arrowheads since the two colors can already indicate directionality.
3. Figure 5b: should choose a different color scheme to better visualize cell types and compositions
4. Figure 5e: the legend is missing
5. Figure 6a,b: The two matrices seem to have the same sets of ligands and receptors on rows and columns. The authors may consider merging the two matrices into one with different markings to separate FB and Endo instead of duplicating the axes.
6. In Line 411, the authors mentioned: "receptors that promote the proliferation and differentiation of TSKs". Any GO analysis? Those receptors contributing to proliferation and differentiation should be highlighted with asterisks in Figure 6b.
7. Line 659-661 is really confusing in terms of the definition of a,b and c.
8. Supplementary Figure 2d legend: "umber" should be "Number"
9. Supplementary Figure 4b-d: the color scheme makes the scores very difficult to see

Response to Reviewers

Overview of Changes

We greatly appreciate the reviewers' concerns and insightful feedback. Our work has been much improved due to their valuable suggestions. We attempted to address every concern by either making appropriate changes to our work or by providing a thorough explanation for our decision. The most significant changes to the manuscript are as follows:

- We have additionally evaluated the performance of Cell2location, CellPhoneDB, and CellChat over the benchmarked datasets, wherein SpaTalk outperforms these methods, consistent with the previous findings.
- We have optimized the algorithms of generating the spatial location of cells by letting the neighbor spots help decide alpha and theta, and added the details about this step.
- We have benchmarked our SpaTalk with other methods with the same LRI database, i.e., CellTalkDB, CytoTalkDB, CellCallDB, CellChatDB, and CellPhoneDB. Concordantly, the superior performance of SpaTalk was observed.
- We have compared computational time between SpaTalk and other methods, wherein the computation time of SpaTalk was within minutes for the decomposition step and the inferring cell-cell communication step, outperforming most deconvolution and signaling inference methods.
- We have added details about the calculation of inter-cellular and intra-cellular scores, and several parameters and functions allowing users to select deconvolution methods, to change the default number of expected cells in each spot, and to show the arrows or not for plotting spatial LRI pairs.

- We have selected the kidney tissue as another application of SpaTalk in replacement of the brain tissue, as suggested by Reviewer#2. Consistent with the previous findings in literature, SpaTalk identified the known intraglomerular communications.
- We have demonstrated the robustness of SpaTalk for signaling inference with the different Slides of the human and mouse kidney tissue (Slide-seq v2), and different patients of SCC (10X Visium).

Please find detailed responses to each concern below.

Reviewer #1

Major comments:

- I was a bit confused with the description about the KNN distance graph. Since the interaction score here is calculated only based on the distance between two cell types from what I understood, this would be the same score for all ligand-receptor pairs between two cell types (sender and receiver). It could be better explained that the inter-cellular score is only calculated based on distance and is not per ligand-receptor pair, but per sender-receiver cell type pair. This is very similar to what is implemented in Giotto and should be acknowledged.

Response: Thanks for your careful review. We are sorry that the previous description of SpaTalk algorithm makes you misunderstand the inter-cellular scores, which are actually different for all ligand-receptor pairs between two cell types (sender and receiver) in SpaTalk. For a given ligand i of the sender (cell type A) and a given receptor j of the receiver (cell type B), the number of ligand-receptor co-expressed cell-cell pairs ($C_{Ai,Bj}$) was obtained from the graph network by counting the 1-hop neighbor nodes of receivers for each sender, resulting in the different number of cell-cell pairs for a given LRI pair between the sender cell type A and the receiver cell type B. In other words, assuming that there is a total of $C_{A,B}$ cell-cell pairs between the sender cell type A and the receiver cell type B, the $C_{Ai,Bj}$ depends on the number of co-

expressed LRI among the $C_{A,B}$ cell-cell pairs and it must be less than $C_{A,B}$ as shown in **Fig. 1** below. We have revised the Methods and Results sections about this part as described above, and we have also revised the Fig. 1c about the process of '(i) Inter-cellular score' in the revised manuscript, in order to avoid the misunderstanding.

Indeed, we are inspired by the implementation in Giotto when developing the SpaTalk method, hence we have added the 'inspired by Giotto' in the corresponding Methods and Results section about the incorporation of KNN distance graph to calculate the inter-cellular score. Besides, we have acknowledged the Giotto in Acknowledgements section in the revised manuscript.

Fig. 1 Process of calculating the inter-cellular and intra-cellular scores with KNN and knowledge graph from the single-cell ST data.

- It was not quite clear to me how spatial coordinates are assigned to each cell type within a spot, could you please explain this part in more details? What does this mean for the knn graph, I saw that you use K=10 in your code, if you consider 30 cells in each spot, does that mean that some of them are directly connected between each other in the knn graph and other not, and how are they connected with cells from other spots? Related to this, how is the downstream signalling implemented for spot-based data where there are multiple different cell types within a spot, which would lead to multiple signalling pathways and targets activated for each of those different cell types? How do the results compare when used on scRNA-seq data instead of spatial data without considering spatial proximity?

Response: Thanks for your comment. As you know, inference of cell-cell communications for spot-based ST data requires the single-cell profiling and

coordinates as the input. However, current deconvolution methods can only dissect the percentage of various cell types. Thus, we proposed the step of spatial mapping to realize the reconstruction of spatial transcriptomics at single-cell resolution. In order to assign a coordinate (\hat{x}, \hat{y}) for each mapped cell in the given spot (x_0, y_0) , we applied following equation to locate the cell into the space:

$$\begin{aligned}\hat{x} &= x_0 + \alpha d_{min} \cos(\theta\pi/180) / 2 \\ \hat{y} &= y_0 + \alpha d_{min} \sin(\theta\pi/180) / 2\end{aligned}$$

where d_{min} represents the spatial distance of the closest neighbor spot, and $\alpha \in (0, 1]$ and $\theta \in (0, 360]$ mean the weight for d_{min} and the angle towards the spot center (x_0, y_0) , respectively.

Fig. 2 Schematic diagram for the generation of spatial coordinates (a) and KNN distance graph (b).

As shown in **Fig. 2a** above, each sampled cell was distributed around the spot center. In the previous version, a random α and θ were used, which might increase the noise of the data as stated by Review #2. Therefore, we proposed a probabilistic distribution for a given cell in each spot (x_0, y_0) by considering the ratio (R) of the same cell type in Q neighbor spots as the probability to locate the cell into the space instead of the random distribution, wherein the θ were first determined by the following probability equation:

$$\hat{P}(\theta) = \frac{R_q + 1}{\sum_{i=1}^Q (R_i + 1)}, \theta \in (90q - 90, 90q]$$

where q represents the q_{th} neighbor spot in Q , which was set to 4 in practice denoting that the space centered the spot were split into four areas and the nearest neighbor in each area was filtered. After determining the θ , the corresponding neighbor spot (x_θ, y_θ) was selected to determine the probabilistic distribution of α

by the following equation:

$$\hat{P}(\alpha) = \begin{cases} (R_{x_0, y_0} + 1)/(R_{x_0, y_0} + R_{x_\theta, y_\theta} + 2), \alpha \in (0, 0.5] \\ (R_{x_\theta, y_\theta} + 1)/(R_{x_0, y_0} + R_{x_\theta, y_\theta} + 2), \alpha \in (0.5, 1] \end{cases}$$

where R_{x_0, y_0} and R_{x_θ, y_θ} represent the ratio of the given cell type in each spot and its neighbor spot.

Given the KNN graph, your understanding is right that some of them are directly connected between each other in the KNN graph and other not in each spot with 30 cells. However, because the generated cells are distributed around the spot center, some cells distributed in the border of the neighbor spots might also be connected as the condition shown in **Fig. 2b** above. We have added the details about how spatial coordinates are assigned to each cell type within a spot to the Methods section as described above and revised the corresponding Result section in the revised manuscript.

Indeed, we agree with you that there are multiple different cell types within a spot and multiple signaling pathways as well as targets activated for each of those different cell types as you mentioned. For spot-based ST data, we first transformed it into single-cell ST data and find the significantly proximal LR pairs between senders and receivers. Then, our SpaTalk filters the LR pairs with known activated downstream signals which are successfully propagated from the receptor to its downstream transcriptional factor (TF) and its target gene in the receiver cell type, because the LRI that mediates the cell-cell communication is supposed to activate at least one TF and its target gene in the receiver cell type as the principle summarized in the recent review (*Nat Rev Genet, 2021*). In detail, we first use knowledge graph to model the ligand-receptor-target (LRT) signaling network with the known regulatory relationship from KEGG and Reactome, then we use a random walk algorithm to determine and score the activated TF and its target gene for a given ligand-receptor pair by using the gene expressed percent among cells of the receiver cell type as the weight of edges in the LRT knowledge graph. As the goal of SpaTalk is to filter and score the LR pairs that mediate the spatially proximal cell-cell communication, the downstream co-expressed TFs and target genes are both used to calculate the final score of each inferred LR pair

(Fig. 1 above), and the greater number of co-expressed TFs and their target genes will lead to a higher score for a given LR pair between the sender cell type and the receiver cell type. We have supplemented the details about the intracellular signal propagation process as described above and revised the Fig. 1c about the process of '(ii) Intra-cellular score' in the revised manuscript, in order to make it clear.

In terms of the comparison of the inferred cell-cell communication results on scRNA-seq data without spatial coordinates, it is hard to evaluate the results for the methods that only consider the ligands and receptors. However, some methods like NicheNet, CytoTalk, and CellCall incorporate the downstream targets to infer the cell-cell communications, which not only infer the significantly enriched LR pairs but also the downstream activated targets triggered by the LRI. For these methods, the result can be compared based on the principle summarized in the recent review (*Nat Rev Genet, 2021*) that the LRI mediating cell-cell communications is supposed to activate the downstream signal pathways, TFs, and their targets. Therefore, we reasoned that a more accurate method would be more likely to enrich the receptor-related biological processes or pathways using the inferred downstream target genes in the receiver cell type, which can be applied to compare the results when used on scRNA-seq data instead of spatial data without considering spatial proximity as you stated. As shown in Fig. 3 below, we benchmarked SpaTalk with NicheNet, CytoTalk, and CellCall using the evaluation index of significance of the enriched ligand-receptor-related pathways based on the inferred targets in the receiver cell type with the Fisher's exact test.

Fig. 3 Schematic illustration for compare the result on scRNA-seq data (left) and the performance comparison of SpaTalk with NicheNet, CytoTalk, and CellCall over the STARmap and seqFISH+ datasets without considering the spatial proximity (right).

- It would increase the usability of the method if there is an option for the user to use a method for deconvolution of ST data of their choice and then only use the second component for inference of cell-cell communication.

Response: Thanks for your valuable comment. As you suggested, we have added two new parameters named 'method' and 'dec_result' in the 'dec_celltype()' function for users to select deconvolution methods including the NNLM of SpaTalk, RCTD, Seurat, SPOTlight, deconvSeq, Stereoscope, and cell2location, or directly use the deconvolution results from other upcoming methods (**Fig. 4** below), followed by the second component for inference of cell-cell communication. We have updated the codes and documents on GitHub, and added a tutorial of this part in the Wiki page.

method	1 means using the SpaTalk deconvolution method, 2 means using RCTD, 3 means using Seurat, 4 means using SPOTlight, 5 means using deconvSeq, 6 means using stereoscope, 7 means using cell2location
env	When method set to 6, namely use stereoscope python package to deconvolute, please define the python environment of installed stereoscope. Default is the 'base' environment. Anaconda is recommended.
anaconda_path	When use python package, please define the path to anaconda, default is ~/anaconda3
dec_result	A matrix of deconvolution result from other upcoming methods, row represents spots or cells, column represents cell types of scRNA-seq reference. See demo_dec_result

Fig. 4 New parameters in the 'dec_celltype()' function.

- Is there an option to change the default number of expected cells in each spot from 30 to another number? The user should be able to change this depending on previous knowledge or whether they can use segmentation to estimate the number.

Response: Thanks for your helpful suggestion and we agree with you that the user should be able to change the estimated cell number in each spot. As you suggested, we have provided a parameter named 'spot_max_cell' for users to change the default number of expected cells in each spot when creating the SpaTalk object (**Fig. 5** below).

```
createSpaTalk(st_data, st_meta, species, if_st_is_sc, spot_max_cell)
```

Arguments	
st_data	A data frame or matrix or dgCMMatrix containing counts of spatial transcriptomics, each column representing a spot or a cell, each row representing a gene.
st_meta	A data frame containing coordinate of spatial transcriptomics with three columns, namely 'spot', 'x', 'y' for spot-based spatial transcriptomics data or 'cell', 'x', 'y' for single-cell spatial transcriptomics data.
species	A character meaning species of the spatial transcriptomics data. 'Human' or 'Mouse'.
if_st_is_sc	A logical meaning if it is single-cell spatial transcriptomics data. TRUE is FALSE.
spot_max_cell	A integer meaning max cell number for each plot to predict. If if_st_sc is FALSE, please determine the spot_max_cell. For 10X (55um), we recommend 30. For Slide-seq, we recommend 1.

Fig. 5 Parameters in the 'createSpaTalk()' function.

Set the expected cell in SpaTalk object

Usage

```
set_expected_cell(object, value)
```

Arguments	
object	SpaTalk object
value	Th number of expected cell for each spot, must be equal to the spot number.

Fig. 6 A new function named 'set_expected_cell ()'.

In addition, we have added a new function named 'set_expected_cell()' to define the variable number of cells in each spot given the situation as you stated (**Fig. 6** above). We have updated the codes and documents on GitHub, and added a tutorial of this part in the Wiki page.

- Was the same LR database (CytoTalkDB) used for comparison of the performance of different cell-cell communication platforms? I could not find this information in the Methods. Otherwise, it would be difficult to compare the results.

Response: Thanks for your comment and you raise an important point. As you suggested, we have benchmarked our SpaTalk with other methods with the same LRI database, i.e., CellTalkDB, CytoTalkDB, CellCallDB, CellChatDB, and CellPhoneDB. As shown in **Fig. 7** below, SpaTalk obtained the most times of the first place across the benchmarked datasets and the underlying LRI databases, outperforming other existing methods for inference of spatially proximal LR pairs that mediating cell-cell communication in space.

Fig. 7 Superior performance of SpaTalk over existing methods. The asterisk represents the top-ranked method for each used LRI database. The times of the first place was labelled beside each method.

However, it is worth to point that several methods also showed decent performance on some individual LRI databases. For example, Giotto obtained the highest median $-\log_{10}P$ among all methods over the STARmap dataset based on CellPhoneDB and CellCallDB, while CytoTalk is the top-ranked method over the STARmap dataset based on CellPhoneDB and CellChatDB considering the median co-expression percent. Over the seqFISH+ OB dataset based on CellChatDB, both of CellPhoneDB and CellChatDB perfectly identified the significantly proximal LRIs in space. For SpaOTsc, it exhibited the highest median co-expression percent over the seqFISH+ SVZ dataset across all LRI databases. We have added the comparison as

described above to the Result section and revised the corresponding Methods section about details of the process of comparison in the revised manuscript.

- Cell2location was not used in the benchmarking for the deconvolution step.

Response: Thanks for your comment. As you suggested, we have also compared the performance of Cell2location in our benchmark. Consequently, although the majority of existing cell-type deconvolution methods can achieve a decent correlation coefficient and low RMSE on spot deconvolution, SpaTalk outperformed these methods on most benchmark datasets with the top-ranked performance (**Fig. 8** below), except for the evaluation indices on the MERFISH dataset and the mean RMSE on the STARmap dataset, which is consistent with the previous conclusion. We have updated the corresponding Result section and figures in the revised manuscript.

Fig. 8 Performance comparison of SpaTalk with other existing cell-type deconvolution methods (RCTD, Seurat, SPOTlight, deconvSeq, Stereoscope, Cell2location). The asterisk represents the top-ranked method for each dataset. NA, not available.

Reviewer #2

Major concerns:

1. In Figure 3, the authors used STARmap brain data to infer short-range signaling between neurons and other cells based on the locations of cell bodies captured by STARmap. However, due to the shape of neurons, interactions can usually happen far away from the cell bodies through axons and synapses that current segmentation methods cannot well capture. The authors should address that point or may consider focusing on other tissues.

Response: Thanks for your comment and we agree with you that interactions can usually happen far away from the cell bodies through axons and synapses due to the shape of neurons. As the current segmentation methods only provide the coordinate of the cell location, it is hard to capture these types of interactions for neurons. As you suggested, we have selected the kidney tissue as another application of SpaTalk in replacement of the brain tissue. Consistent with the previous findings in literature, SpaTalk identified the intraglomerular communications among glomerular cells in kidney as shown in **Fig. 1** below. We have revised the Result section in the revised manuscript as follows:

“Identification of signal transmission among glomerular cells in kidney. SpaTalk was then applied to investigate and visualize the intraglomerular communications over the Slide-seq ST dataset (v2) of the mouse kidney (Fig. 1a), including data of 20591 sequenced genes for 27044 spots in space covering the spatial axis of collecting duct intercalated cells (CD-IC), collecting duct principal cells (CD-PC), distal convoluted tubules (DCT), endothelial cells (Endo), fibroblasts (FB), granular cells (GC), macrophages (Macro), mesangial cells (MC), immune cells, proximal convoluted tubules (PCT), podocytes (Pod), thick ascending limb (TAL), and vascular smooth muscle cells (vSMC) from cortex of kidney to renal medulla. As shown in Fig. 1b, SpaTalk identified the spatial signal transmission among Pod, MC, and Endo in glomerulus. For example, the direct cell-cell communication mediated by the pleiotrophin (Ptn) and protein tyrosine phosphatase receptor type B (*Ptprb*)

interaction was observed from MC and Pod to Endo, wherein Ptn is a secreted growth factor that can bind *Ptprb*, which is known to be involved with adherens junction stimulating endothelial cell migration and maintaining proper glomerular function. Besides, collagen and Notch signaling were also identified forming the matrix that provides structural support for the glomerulus, which are necessary for proper glomerular basement membrane formation and glomerular development. Concordantly, these identified LRIs among glomerular cells are associated with multiple biological processes and pathways that play vital roles in the regulation of physiological kidney development and glomerular filtration function in the urinary system, including tube morphogenesis, positive regulation of cell adhesion, urogenital system development, and morphogenesis of a branching epithelium (Fig. 1c).

Notably, the Pod-Endo communication mediated by vascular endothelial growth factor a (*Vegfa*) signaling was significantly enriched in space, in accordance with the fact that *Vegfa* is vital for the formation and maintenance of select microvascular beds within the kidney. It is known that *Vegfa*, normally produced by healthy podocytes, has been shown to be a critical regulator of glomerular development and function and precise expression of the amount of *Vegfa* is required for adequate barrier function. As the kinase insert domain receptor (*Kdr*) of *Vegfa*, *Kdr* functions as the main mediator of VEGF-induced endothelial proliferation, survival, migration, tubular morphogenesis and sprouting. Concordantly, the Pod-Endo communication mediated by *Vegfa-Kdr* was also identified in other mouse kidney ST data. In addition, most shared LRIs mediating the intraglomerular communications in space were also observed across slides of human and mouse kidney, suggesting the robustness and universality of the spatially resolved cell-cell communications inferred by SpaTalk.

We then applied SpaTalk to another spot-based ST dataset of the mouse kidney (10X Visium), reaching up to 19,465 unique genes among 3,124 spots in space (Fig. 1f). Leveraging previously published adult mouse kidney cell taxonomy by single-nucleus RNA-seq data (Supplementary Fig. 5c), SpaTalk reconstructed the spatial transcriptomics atlas at the single-cell resolution for Slide-seq data, which showed consistent spatial localization of glomerular and other cells (Fig. 1f).

LHAL, loop of Henle ascending loop. **g** Significantly enriched LRIs that mediate Pod-Endo communications. Top 20 LRIs (left) and spatial distribution of the *Angpt1-Tek* pairs between the Pod senders and Endo receivers were plotted. **h** Communications of Endo-MC mediated by the *Pdgfb-Pdgfrb* interaction in space and the spatial distances of *Pdgfb-Pdgfrb* in Endo-MC and all cell-cell pairs, respectively. *P* values were calculated with the Wilcoxon test.

Compared to Slide-seq (v2) data, loop of Henle descending loop (LHDL) and loop of Henle ascending loop (LHAL) were also observed in the 10X Visium data. Consistently, most LRIs mediating intraglomerular communications in Slide-seq (v2) data were also found in 10X Visium data (Fig. 1g). Similar to the *Vegfa* paracrine system, Angiopietin-1 (*Angpt1*) is expressed by podocytes and its cognate tyrosine kinase receptor (*Tek*) is expressed by the glomerular endothelial cells, which plays an indispensable role in glomerular health and maintenance of the filtration barrier in physiological conditions. Moreover, the direct communication between Endo to MC mediated by platelet-derived growth factor B and its receptor (*Pdgfb-Pdgfrb*) was significantly enriched (Fig. 1h), in accordance with the classical studies that have delineated a key role for the *Pdgfb* in communication between the glomerular endothelium and nearby mesangial cells.”

2. The authors compared their method against multiple decomposition and cell-cell interaction algorithms. However, cell2location (decomposition), cellphoneDB (interaction) and cellchat(interaction) have been missed. These packages have been widely used. It would be more convincing to benchmark against them, especially cell2location.

Response: Thanks for your comment. As you suggested, we have also compared the performance of Cell2location in our benchmark. Consequently, although the majority of existing cell-type deconvolution methods (RCTD, Seurat, SPOTlight, deconvSeq, Stereoscope, and Cell2locatoion) can achieve a decent correlation coefficient and low RMSE on spot deconvolution, SpaTalk outperformed these methods on most benchmark datasets with the top-ranked performance (Fig. 2a-b below), except for the evaluation indices on the MERFISH dataset and the mean RMSE on the STARmap dataset, which is consistent with the previous conclusion.

Fig. 2 Comparison of SpaTalk with other methods. **a** Schematic diagram for generating simulated spot data. Cells were split according to the fixed spatial distance and then merged for the single-cell ST data with known cell types. **b** Performance comparison of SpaTalk with other existing cell-type deconvolution methods (RCTD, Seurat, SPOTlight, deconvSeq, Stereoscope, Cell2location). The asterisk represents the top-ranked method for each dataset. NA, not available. **c** Schematic illustration of the procedure and rationale for single-cell ST data to evaluate predicted LRIs that mediate spatially resolved cell–cell communications. **d** and **e** Performance comparison of SpaTalk with existing cell–cell communication inference methods (Giotto, SpaOTsc, NicheNet, CytoTalk, CellCall, CellPhoneDB, and CellChat) on the STARmap and seqFISH+ datasets. The P value represents the difference of spatial distances between sender–receiver and all cell–cell pairs assessed with the Wilcoxon test.

In terms of the CellPhoneDB and CellChat, we have also evaluated the performance of them over the benchmarked datasets. As shown in **Fig. 2c-d** above, although most LRIs inferred by other methods showed significantly closer spatial distances between sender–receiver pairs than that between all cell–cell pairs, superior performance of SpaTalk was observed, ranking first for both evaluation indices for STARmap datasets. Similarly, SpaTalk obtained a higher median $-\log_{10}P$ value and co-expression percent on the seqFISH+ OB and SVZ datasets but not for SpaOTsc, CellPhoneDB, and CellChat on individual evaluation indices (**Fig. 2e** above), which is consistent with the previous conclusion. We have revised the corresponding Result section by additionally evaluating the performance of Cell2location, CellPhoneDB, and CellChat as described above in the revised manuscript.

3. The authors should also compare computational time between SpaTalk and other methods.

Response: Thanks for your comment. As you suggested, we have compared computational time between SpaTalk and other methods in terms of the decomposition and the inferring cell-cell communication steps. For the decomposition step, SpaTalk and other deconvolution six methods were compared over four simulated (STARmap, MERFISH, seqFISH+ OB and SVZ) and one real spot-based (10X Visium) ST datasets, wherein the computation time of SpaTalk was within minutes similar to RCTD and Seurat, outperforming SPOTlight, deconvSeq, Stereoscope, and Cell2location (**Table 1** below).

Table 1. Computation time of the deconvolution step.

Methods	Simulated spot data				Real spot data
	STARmap	MERFISH	seqFISH OB	seqFISH SVZ	10X Visium
SpaTalk	0.13min	0.08min	2.18min	3.05min	6.23min
RCTD	1.35min	0.65min	1.20min	3.80min	3.33min
Seurat	0.22min	0.18min	0.90min	0.55min	2.47min
SPOTlight	1.68min	0.32min	4.30min	10.92min	26.83min
deconvSeq	2.18min	NA	25.85min	17.52min	NA
Stereoscope	56.58min	205.53min	359.20min	67.02min	604.58min
Cell2location	35.13min	20.30min	25.25min	30.02min	31.03min

NA, not available.

Table 2. Computation time of the inferring cell-cell communication step.

Methods	STARmap	seqFISH OB (FOV0)	seqFISH SVZ(FOV0)	10X Visium (TSK-Endo)
SpaTalk	0.47min	3.40min	22.05min	4.72min
Giotto	5.02min	5.06min	3.99min	37.11min
SpaOTsc	0.08min	0.09min	0.11min	26.70min
NicheNet	19.39min	13.09min	11.00min	4.59min
CytoTalk	42.03min	>12h	>12h	123.68min
CellCall	2.34min	12.66min	16.26min	14.86min
CellPhoneDB	1.57min	1.67min	3.69min	11.30min
CellChat	0.69min	67.65min	123.90min	41.48min

For the inferring cell-cell communication step, SpaTalk and other seven methods were benchmarked over three single-cell ST datasets (STARmap, seqFISH+ OB and SVZ)

and one reconstructed single-cell ST dataset (10X Visium) with SpaTalk. As shown in **Table 2** above, the computation time of SpaTalk, Giotto, SpaOTsc, NicheNet, CellCall, and CellPhoneDB were all within minutes, superior to that of CytoTalk and CellChat. We have added the comparison to the Result section as described above in the revised manuscript.

4. For spot-based spatial data, the authors introduced arbitrarily generated locations of single-cells (x,y) around each spot (x_0,y_0) (Figure 6c, Line 598-600) which increases the noise of the data. Is it a must? Could neighbor spots around each spot also be used to help decide alpha and theta?

Response: Thanks for your helpful suggestion and we totally agree with your point. In our pipeline, it is a must to generated locations of single cells in each spot sinc it is the foundation for the subsequent inference of spatially resolved cell-cell communications over the spot-based ST data. However, it is indeed not a must to arbitrarily generate locations, which might increase the noise of the data as you stated. Inspired by your idea, we reason that the higher ratio of the same cell type in the neighbor spot will lead to the closer distance for a given cell towards the corresponding neighbor spot, namely letting the neighbor spots help decide alpha and theta as you suggested. To be more specific, we have proposed a probabilistic distribution for a given cell in each spot (x_0, y_0) by considering the ratio (R) of the same cell type in Q neighbor spots as the probability to assign a coordinate (\hat{x}, \hat{y}) for each sampled cell with the following function:

$$\begin{aligned}\hat{x} &= x_0 + \alpha d_{min} \cos(\theta\pi/180)/2 \\ \hat{y} &= y_0 + \alpha d_{min} \sin(\theta\pi/180)/2\end{aligned}$$

where d_{min} represents the spatial distance of the closest neighbor spot, and $\alpha \in (0, 1]$ and $\theta \in (0, 360]$ mean the weight for d_{min} and the angle towards the spot center (x_0, y_0) , respectively. In detail, the θ were first determined by the following probability equation:

$$\hat{P}(\theta) = \frac{R_q + 1}{\sum_{i=1}^Q (R_i + 1)}, \theta \in (90q - 90, 90q]$$

where q represents the q_{th} neighbor spot in Q , which was set to 4 in practice

denoting that the space centered the spot were split into four areas and the nearest neighbor in each area was filtered. After determining the θ , the corresponding neighbor spot (x_θ, y_θ) was selected to determine the probabilistic distribution of α by the following equation:

$$\hat{P}(\alpha) = \begin{cases} (R_{x_0, y_0} + 1)/(R_{x_0, y_0} + R_{x_\theta, y_\theta} + 2), \alpha \in (0, 0.5] \\ (R_{x_\theta, y_\theta} + 1)/(R_{x_0, y_0} + R_{x_\theta, y_\theta} + 2), \alpha \in (0.5, 1] \end{cases}$$

where R_{x_0, y_0} and R_{x_θ, y_θ} represent the ratio of the given cell type in each spot and its neighbor spot.

We have revised the Method section as described above in the revised manuscript and updated the corresponding codes and documents on GitHub.

5. Handling biological replicates is a key challenge for signaling inference that spatial algorithms start to meet (One example is the `multinichenetr` algorithm being developed <https://github.com/saeyslab/multinichenetr>). How would the cell decomposition step and the graph learning step handle replicates and how robust will it be?

Response: Thanks for your comment and you raise an important point. We agree with you that handling biological replicates has been always the key challenge for signaling inference methods including the single-cell algorithms over scRNA-seq data and the spatial algorithms over ST data with growing concern. To make the signaling inference as replicable as possible, we used the normalized expression of all detected genes instead of the count and partial genes, e.g., differentially expressed genes (DEGs) or marker genes, to dissect the cell-type composition, in order to mitigate the influence of highly variable genes and different platforms, individuals, and batches in the cell decomposition step. Given the inference of spatially resolved cell-cell communication mediated by ligand-receptor interactions, we used the co-expression of ligands and receptors over the KNN distance graph instead of the significantly high expression of ligands/receptors to mitigate the influence of variably expressed ligands and receptors. In the same way, we used the co-expression of receptors, transcriptional factors (TFs), and target genes among all receiver cells instead of using the absolute expression to mitigate the influence of variably expressed TFs and their target genes in the graph

learning step.

As you suggested, we have evaluated the robustness of SpaTalk for signaling inference. First, we applied SpaTalk to the human and mouse kidney tissues with multiple slides sequenced by Slide-seq v2 (*iScience*, 2022), wherein we selected slides with at least 15 MCs to infer the cell-cell communications. As shown in **Fig. 3** below, several known LRIs that have been widely reported to mediate the intraglomerular communications for maintaining proper glomerular function were identified such as VEGFA and PTN signaling, which were shared in most slides. Although the total number of inferred LRI pairs were different across nine slides, more than half of significantly enriched LRI pairs for each slide were existed in other slides with the median percent of shared LRI pairs reaching up to 75% and 65% for human and mouse slides, respectively.

Fig. 3 Cell-cell communication inference among the glomerular cells in kidney by SpaTalk. Inferred LRI pairs mediating the intraglomerular communications across different slides of human and mouse kidney sequenced by Slide-seq (v2). Shared LR pairs among different slides were plotted.

In addition, we compared the significantly enriched LRI pairs mediating the stroma-TSK communications of human SCC ST datasets from two patients, namely P. 2 and P. 10 as shown in **Fig. 4** below. Concordantly, a substantial shared LRI pairs and downstream TFs were identified between P. 2 and P. 10. For P. 2, the median percent of shared LRI pairs and TFs reach 82% and 77%, respectively. For P. 10, the percent of shared LRI pairs and TFs range from 86% to 96% and from 93% to 99%, respectively. The present results indicate that SpaTalk is a relatively robust method with high

biological replicability for signaling inference across different slides and individuals. We have also revised the Resection about the two applications of SpaTalk as described above in the revised manuscript.

Fig. 4 Comparison of inferred LRI pairs and TFs by SpaTalk over 10X Visium ST datasets from two SCC patients (P. 2 and P. 10). The percent of shared LRI pairs and TFs for each patient was labeled beside the total number of inferred LRI pairs and TFs.

Minor:

1. Fig 1c: “siganling” should be “signaling”

Response: Thanks for your careful review. We have revised the typo of Fig. 1c in the revised manuscript.

2. For visualizing cell-cell crosstalks, the arrows on top of cells are very difficult to see (such as Fig 4g, 6g, and supplementary Fig 6). The authors can consider just using lines without arrowheads since the two colors can already indicate directionality.

Response: Thanks for your useful suggestion. As you suggested, we have revised the visualization of cell-cell crosstalks by using lines instead of arrowheads in Fig.4g, 6g, and supplementary Fig. 6, e.g., **Fig. 5** below. Besides, we have added a new parameter named ‘if_show_arrow’ in the plotting functions allowing users to show the directionality or not inspired by your suggestion. We have also updated the code and document on Github and the Wiki page.

Fig. 5 Communications from EMT-like and EMT-unlike TSKs to CAFs mediated by the MMP1–CD44 interaction in space.

3. Figure 5b: should choose a different color scheme to better visualize cell types and compositions

Response: Thanks for your comment. We have changed a different color scheme to visualize cell types and compositions (**Fig. 6** below) and updated the corresponding figure in the revised manuscript.

Fig. 6 Spatial characterization of tumor and stromal cells in human squamous cell carcinoma Visium data with SpaTalk. **a** Visium spot-based ST dataset of human skin SCC in patient 2 with the matched scRNA-seq dataset involving the main keratinocytes (KC), stromal cells, and immune cells. **b** Cell-type decomposition by SpaTalk. Cyc, cycling; Diff, differentiating; NK, natural killer; FB, fibroblast. **c** TSK percent and TSK score across spatial spots. The expression of known TSK markers is plotted. **d** Pearson's

correlation coefficient between the TSK percent and TSK score. **e** Cell-type decomposition by SpaTalk at single-cell resolution for the spot-based human skin SCC ST data. **f** Contour plot of TSK, FB, and Endo based on the reconstructed single-cell ST atlas by SpaTalk.

4. Figure 5e: the legend is missing

Response: Thanks for your careful review. We have added the legend for Fig. 5e in the revised manuscript as follows: **e** Cell-type decomposition by SpaTalk at single-cell resolution for the spot-based human skin SCC ST data. (**Fig. 6** above)

5. Figure 6a,b: The two matrices seem to have the same sets of ligands and receptors on rows and columns. The authors may consider merging the two matrices into one with different markings to separate FB and Endo instead of duplicating the axes.

Response: Thanks for your comment. As you suggested, we have revised the Fig. 6a and 6b by merging the two matrices into one with different markings to separate FB and Endo instead of duplicating the axes (**Fig. 7** below) and updated the corresponding figure in the revised manuscript.

Fig. 7 Top 20 inferred LRIs that mediate cell–cell communication from the TSK senders to the FB and Endo receivers and from the FB and Endo senders to TSK receivers. The number sign represents the receptor contributing to the proliferation and differentiation of TSK.

6. In Line 411, the authors mentioned: “receptors that promote the proliferation and differentiation of TSKs”. Any GO analysis? Those receptors contributing to proliferation and differentiation should be highlighted with asterisks in Figure 6b.

Response: Thanks for your comment. As you suggested, we have supplemented GO analysis on the TSK receptors that promote the proliferation and differentiation of TSKs

triggered by the MDK, HGF, HMGB1, and THBS1 ligands (**Table 3** below). From the GO analysis, the PTPRZ1, SDC1, and ITGA6 are highly related with the cell development, differentiation, and migration, consistent with the characteristics of TSK. Also, we have highlighted those receptors contributing to proliferation and differentiation with number sign. We have updated the corresponding figure and figure legend in the revised manuscript.

Table 3 GO analysis of TSK receptors

Gene	GO_ID	GO_term
PTPRZ1	GO:0007417	central nervous system development
PTPRZ1	GO:0002244	hematopoietic progenitor cell differentiation
PTPRZ1	GO:0048709	oligodendrocyte differentiation
PTPRZ1	GO:0048714	positive regulation of oligodendrocyte differentiation
PTPRZ1	GO:0070445	regulation of oligodendrocyte progenitor proliferation
SDC1	GO:0060070	canonical Wnt signaling pathway
SDC1	GO:0016477	cell migration
SDC1	GO:0050900	leukocyte migration
SDC1	GO:0048627	myoblast development
SDC1	GO:0060009	Sertoli cell development
SDC1	GO:0055002	striated muscle cell development
SDC1	GO:0001657	ureteric bud development
ITGA6	GO:0098609	cell-cell adhesion
ITGA6	GO:0007160	cell-matrix adhesion
ITGA6	GO:0050900	leukocyte migration
ITGA6	GO:0050873	brown fat cell differentiation
ITGA6	GO:0010668	ectodermal cell differentiation
ITGA6	GO:0030198	extracellular matrix organization
ITGA6	GO:0022409	positive regulation of cell-cell adhesion
ITGA6	GO:0030335	positive regulation of cell migration
ITGA6	GO:0035878	nail development

7. Line 659-661 is really confusing in terms of the definition of a , b and c .

Response: Thanks for your comment. We have added a more clear description and a table (**Table 4** below) about the pathway enrichment analysis with Fisher's exact test, wherein a is the number of inferred target genes (interested genes) that match a given pathway; b is the number of a given pathway's genes that exclude a , namely the uninterested genes that match a given pathway; c is the number of inferred

target genes (interested genes) that unmatch a given pathway; d is the number of all genes excluding a , b , and c , namely the uninterested genes that unmatch a given pathway. We have revised the Methods section about the pathway enrichment analysis with Fisher's exact test as described above in the revised manuscript.

Table 4. Pathway enrichment analysis with Fisher's exact test

	Interested genes	Uninterested genes
Genes matching the pathway	a	b
Genes unmatching the pathway	c	d

8. Supplementary Figure 2d legend: "umber" should be "Number"

Response: Thanks for your careful review. We have revised the typo in Supplementary Figure 2d legend.

9. Supplementary Figure 4b-d: the color scheme makes the scores very difficult to see

Response: Thanks for your comment. As you suggested, we have tried a different color schemes, but the result seems the same (**Fig. 8** below). We find the reason is that the scores and percent of these non-hepatocyte cell types in most spots are very low, which makes them hard to distinguish from hepatocytes. Besides, we have provided the Source Data as a single Excel file related to this figure.

Fig. 8 Cell-type decomposition on the mouse liver ST data of Slide-seq. **a** Mouse liver scRNA-seq reference integrating the non-parenchymal cells from the mouse cell atlas (MCA) and the parenchymal hepatic cells from GSE12568847, which contains 6,029 cells involving the major immune cells and the pericentral and periportal hepatocytes, etc. **b-d** Expression of known marker gene (up) and the percent (down) for Endo, B cell, DC, Epithelia, Erythroblast, granulocyte, Kupffer cell, Macro, neutrophil, stromal cell, and T cell. **e** Percent of cell types across 25,595 spots of Slide-seq data.

REVIEWERS' COMMENTS

Reviewer #1 (Remarks to the Author):

The authors addressed all of my comments.

Reviewer #2 (Remarks to the Author):

Shao, Li, and Yang et al. have addressed most of my concerns. The manuscript and the algorithm have both improved a lot.

The only minor concern left is the visibility of Supplementary Figure 4. It may be possible to make the signals visible by enlarging the dots with non-zero values and allowing partial overlaps - but that's totally optional.

Point-by-point response to the reviewers' comments

Reviewer #1:

The authors addressed all of my comments.

Response: Thanks for your positive comment.

Reviewer #2

Shao, Li, and Yang et al. have addressed most of my concerns. The manuscript and the algorithm have both improved a lot.

The only minor concern left is the visibility of Supplementary Figure 4. It may be possible to make the signals visible by enlarging the dots with non-zero values and allowing partial overlaps - but that's totally optional.

Response: Thanks for your helpful suggestion. As you suggested, we have made the signals visible by enlarging the dots with non-zero values and allowing partial overlaps and changed a visible color scheme to improve the visibility of Supplementary Figure 4 (see **Fig. 1** below), which was also done to improve the visibility for Figure 5 in the revised manuscript.

Fig. 1 Cell-type decomposition on the mouse liver ST data of Slide-seq. **a** Mouse liver scRNA-seq reference integrating the non-parenchymal cells from the mouse cell atlas (MCA) and the parenchymal hepatic cells from GSE125688, which contains 6,029 cells involving the major immune cells and the pericentral and periportal hepatocytes, etc. **b-d** Expression of known marker gene (up) and the percent (down) for Endo, B cell, DC, Epithelia, Erythroblast, granulocyte, Kupffer cell, Macro, neutrophil, stromal cell, and T cell. **e** Percent of cell types across 25,595 spots of Slide-seq data. For the boxplots (minima, 25th percentile, median, 75th percentile, and maxima), the numbers of data points for each box are 25,595. PC, pericentral; PP, periportal; Hep, hepatocytes; Endo, endothelial cells; DC, dendritic cells; Macro, macrophages.